# Approximate Discretization Invariance for Deep Learning on Neural Fields

**Clinton J. Wang**                                                        CLINTONW@CSAIL.MIT.EDU

**Polina Golland**                                                          POLINA@CSAIL.MIT.EDU

*MIT Computer Science & Artificial Intelligence Laboratory, Cambridge MA, USA*

**Editors:** Sophia Sanborn, Christian Shewmake, Simone Azeglio, Arianna Di Bernardo, Nina Miolane

## Abstract

While neural fields have emerged as powerful representations of continuous data, there is a need for neural networks that can perform inference on such data without being sensitive to how the field is sampled, a property called (approximate) discretization invariance. We develop DI-Net, a framework for learning discretization invariant operators on neural fields of any type. Whereas current theoretical analyses of discretization invariant networks are restricted to the limit of infinite samples, our analysis does not require infinite samples and establishes upper bounds on the variation in DI-Net outputs given different finite discretizations. Our framework leads to a family of neural networks driven by numerical integration via quasi-Monte Carlo sampling with discretizations of low discrepancy. DI-Nets manifest desirable theoretical properties such as universal approximation of a large class of maps between $L^2$ functions, and gradients that are also discretization invariant. DI-Nets can also be seen as generalizations of many existing network families as they bridge discrete and continuous network classes, such as convolutional neural networks (CNNs) and neural operators respectively. Experimentally, DI-Nets derived from CNNs are able to classify and segment visual data represented by neural fields under various discretizations, and sometimes even generalize to new types of discretizations at test time.

**Keywords:** Neural fields, discretization invariance, universal approximation

## 1. Introduction

Neural fields (NFs), which encode signals as the parameters of a neural network, have many useful properties. NFs can efficiently store and stream continuous data (Sitzmann et al., 2020b; Dupont et al., 2022; Takikawa et al., 2022), represent and render detailed 3D scenes at lightning speeds (Müller et al., 2022), and integrate data from a wide range of modalities (Gao et al., 2022). NFs are thus an appealing data representation for many applications.

However, current approaches for training networks on a dataset of NFs have major limitations. The sampling-based approach converts such data to pixels or voxels as input to discrete networks (Vora et al., 2021), but it incurs interpolation errors and does not leverage the ability to evaluate the NF anywhere on its domain. The hypernetwork approach trains a model to predict NF parameters (or a lower dimensional "modulation" of such parameters) which can be tailored for downstream tasks (Tancik et al., 2020a; Dupont et al., 2022; Mehta et al., 2021), but hypernetworks based on the parameter space of one type of NF are incompatible with other types. Moreover, hypernetworks are unsuitable for important classes of NFs whose parameters extend beyond a neural network. Our work strengthens the sampling-based approach with the notion of *approximate discretization invariance*: the output of an operator that processes a continuous signal by sampling it at a set of discrete

points should be largely independent of how the sample points are chosen, particularly as the number of points becomes large[1].

In this paper we propose the *DI-Net*, a discretization invariant neural network for learning and inference on NFs. DI-Nets leverage numerical integration to yield desirable convergence properties, can be applied to any type of NF, and are universal approximators on a large class of maps. DI-Nets are a broad class of models that can be applied to classification, segmentation, and many other tasks.

## 2. Principles of DI-Nets

Let $\Omega$ be a bounded measurable subset of a $d$-dimensional compact metric space. Let $\mathcal{F}_c$ denote the space $L^2(\Omega, \mathbb{R}^c) = \{f : \Omega \to \mathbb{R}^c : \int_\Omega \|f\|^2 d\mu < \infty\}$. We refer to all maps $f \in \mathcal{F}_c$ as NFs, although such maps also encompass other types of signals and data formats. We call $d$ the dimensionality of the NF and $c$ the number of channels.[2]

**Definition 1** *A DI-Net layer $\mathcal{H}$ is a bounded continuous map $\mathcal{F}_n \to \mathcal{F}_m$ (e.g., linear combinations of channels), or $\mathbb{R}^n \to \mathcal{F}_m$ (e.g., in a generative model), or $\mathcal{F}_n \to \mathbb{R}^m$ (e.g., in a classifier), or $\mathcal{F}_n \times \mathcal{F}_m \to \mathcal{F}_c$ (e.g., in style transfer) for some $n, m, c \in \mathbb{N}$. If the map's domain is $\mathcal{F}_n$, it must also be discretization invariant (Definition 3). A DI-Net is a directed acyclic graph of such layers.[3]*

**Proposition 2** *A DI-Net is invariant to the parameterization of the NF used to train it. If $f, g \in \mathcal{F}_c$ are equal in the sense that $f_i = g_i$ for all $i \in \{1, ..., c\}$, then any DI-Net or loss function on $\mathcal{F}_c$ maps $f$ and $g$ to the same output.*

This property follows directly from our definition of the DI-Net, and confers interoperability that is missing from hypernetwork or modulation-based learning approaches.

For simplicity we characterize single-channel NFs and DI-Net layers, and in Appendix C we extend our results to the multi-channel case. The *discretization* of a neural field $f \in L^2(\Omega)$ is a point set $X \subset \Omega$ on which the NF is evaluated. We say that the map $\mathcal{H}_\phi : L^2(\Omega) \to \mathbb{R}^n$ is *discretizable* if it induces a map $\mathfrak{H}_{\phi,X} : L^2(\Omega)/I_X \to \mathbb{R}^n$ for every discretization $X$, where $I_X$ is the ideal of $L^2(\Omega)$ that vanishes at each $x \in X$.

**Definition 3** *A discretizable map $\mathcal{H}_\phi : L^2(\Omega) \to \mathbb{R}^n$ is discretization invariant if for every discretization $X$ and input $f$, the deviation of $\mathcal{H}_\phi$ from the induced map $\hat{\mathcal{H}}_\phi^X$ is bounded by the product of the variation of $f$ and the discrepancy of $X$. A map $\bar{\mathcal{H}}_\phi : L^2(\Omega) \to L^2(\Omega)$ is discretization invariant if it is discretization invariant at each point in its image.*

A function's variation measures how much it fluctuates over its domain, and a discretization's discrepancy is low for dense, evenly distributed point sets (see Appendix B for precise definitions). Their product is precisely the upper bound in the celebrated Koksma–Hlawka inequality, which bounds the difference between a function's integral and its sample mean:

---

1. Although the name "discretization invariance" implies strict equality for all discretizations, this term is usually used in the literature to denote this weaker approximation condition, hence we drop the qualifier "approximate" for most of this work.

2. For example, an occupancy network (Mescheder et al., 2018) is 3-dimensional and has 1 channel. NeRF (Mildenhall et al., 2020) is 5-dimensional (world coordinates and view angles) and has 4 channels.

3. Note that many common loss functions also generalize naturally to the continuous domain as bounded continuous maps $\mathcal{F}_c \to \mathbb{R}$ or $\mathcal{F}_c \times \mathcal{F}_c \to \mathbb{R}$.

**Definition 4** *A function $f \in L^2(\Omega)$ satisfies a Koksma–Hlawka inequality if for any point set $X \subset \Omega$,*

$$\left| \frac{1}{N} \sum_{j=1}^{N} f(x') - \int_{\Omega} f(x)\, dx \right| \leq V(f)\, D(X), \tag{1}$$

*for normalized measure $dx$, variation $V$ of the function and discrepancy $D$ of the point set.*

This naturally leads to parameterizations of discretization invariant layers as integrals over $\Omega$. In the $L^2(\Omega) \to \mathbb{R}$ case:

$$\mathcal{H}_\phi[f] = \int_{\Omega} h(x, f(x), \ldots, D^\alpha f(x); \phi) d\mu(x), \tag{2}$$

for a function $h$ which is Fréchet differentiable at all $f \in L^2(\Omega)$, is of bounded variation in $\Omega$ for all $f$, is differentiable w.r.t. its parameters $\phi$, and may depend on weak derivative $D^\alpha f = \frac{\partial^{|\alpha|} f}{\partial x_1^{\alpha_1} \ldots \partial x_n^{\alpha_n}}$ for multi-index $\alpha$.[4] Its induced map under discretization $X$ is then:

$$\hat{\mathcal{H}}_\phi^X[f] = \sum_{x \in X} h(x, f(x), \ldots, D^\alpha f(x); \phi)\, w(x; \mu, X), \tag{3}$$

where $w(x; \mu, X)$ are quadrature weights ($1/|X|$ by default). We can write the $L^2(\Omega) \to L^2(\Omega)$ case similarly. Such a parameterization opens up a rich toolkit of numerical integration methods, although here we focus on quasi-Monte Carlo (QMC) (Caflisch, 1998), which generates low discrepancy discretizations. Our definition of discretization invariance immediately yields a class of discretization sequences under which discretized maps converge:

**Definition 5** *We call a sequence of discretizations $\{X_N\}_{N \in \mathbb{N}}$ whose discrepancy tends to 0 as $N \to \infty$ an equidistributed discretization sequence.[5]*

**Proposition 6** *If $\mathcal{H}_\phi$ is a discretization invariant map and $f \in L^2(\Omega)$ has finite variation, $\lim_{N \to \infty} \hat{\mathcal{H}}_\phi^{X_N}[f] = \mathcal{H}_\phi[f]$ for any equidistributed discretization sequence $\{X_N\}_{N \in \mathbb{N}}$.*

Thus a network composed of discretization invariant layers converges in the forward direction. We can also establish convergence of the network's discretized gradients as follows (proved and expanded on in Appendix C.2):

**Theorem 7** *A DI-Net permits backpropagation of its outputs with respect to its input as well as all its learnable parameters. Under an equidistributed discretization sequence, the gradients of each layer converge to the appropriate derivative under the measure on $\Omega$.*

---

4. The dependence of $h$ on weak derivatives up to order $k = |\alpha|$ requires that they are integrable, i.e. $f$ is in the Sobolev space $W^{k,2}(\Omega)$.

5. Any equidistributed sequence of points generates an equidistributed discretization sequence by truncating to the first $N$ terms, although the class of all equidistributed discretization sequences is much larger than this. Quasi-Monte Carlo sampling can efficiently generate equidistributed sequences on many domains.

Recall that discretization invariant maps were defined in terms of an arbitrary discretizable map on a continuous domain. Then Proposition 6 suggests that for every discretizable map there exists some discretization of sufficiently low discrepancy such that the induced discrete map approximates the continuous-domain map to desired precision for particular types of functions. This universal approximation theorem can be formalized as follows:

**Theorem 8** *Let* $\mathcal{G}, \mathcal{G}'$ *be subsets on* $L^2(\Omega)$ *restricted to piecewise smooth functions, with mild conditions on* $\Omega$. *For every Lipschitz continuous map* $\mathcal{R} : \mathcal{G} \to \mathcal{G}'$, *there exists a DI-Net* $\mathcal{T}$ *that approximates it to arbitrary accuracy w.r.t. a finite measure* $\nu$ *on* $\mathcal{G}$. *As a corollary, every Lipschitz continuous map* $\mathcal{G} \to \mathbb{R}^n$ *or* $\mathbb{R}^n \to \mathcal{G}$ *can also be approximated by some DI-Net.*

## 3. Design and Implementation of DI-Nets

DI-Nets encompass a very large family of neural networks. On one side of the DI-Net family includes networks such as DeepONets and neural operators (Kovachki et al., 2021b; Lu et al., 2021), which can learn general maps between function spaces but in practice are designed to solve partial differential equations. On the other side are networks that are designed entirely on discrete domains such as convolutional neural networks (CNNs), but can be readily extended to the continuous domain. In the same way that neural fields extend signals on point clouds, meshes, grids, and graphs to a compact metric space, DI-Nets extend neural networks that operate on discrete signals by converting every layer to an equivalent discretizable map.

We illustrate the case of CNNs: we describe how to extend convolutional layers to DI-Nets here, and discuss other constructions in Appendix D, including normalization, pooling, upsampling, skip connections, tokenization and attention. The resulting convolutional DI-Nets can be initialized directly with the weights of a pre-trained CNN as we investigate in Appendix G.1.

**Convolution** For a measurable $S \subset \Omega$ and a polynomial basis $\{p_{j,\phi}\}_{j \geq 0}$ that spans $L^2(S)$, $S$ is the support of a polynomial convolutional kernel $K_\phi : \Omega \times \Omega \to \mathbb{R}$ defined by:

$$K_\phi(x, x') = \begin{cases} \sum_{j=0}^n p_{j,\phi}(x - x')^j & \text{if } x - x' \in S \\ 0 & \text{otherwise.} \end{cases} \tag{4}$$

for some chosen $n \in \mathbb{N}$. A convolution is the linear map $\mathcal{H}_\phi : L^2(\Omega) \to L^2(\Omega)$ given by:

$$\mathcal{H}_\phi[f] = \int_\Omega K_\phi(\cdot, x') f(x') dx'. \tag{5}$$

An MLP convolution is defined similarly except the kernel becomes $\tilde{K}_\phi(x, x') = \text{MLP}(x - x'; \phi)$ in the non-zero case. While MLP kernels are favored over polynomial kernels in many applications due to their expressive power (Wang et al., 2021), polynomial bases can be used to construct filters satisfying desired properties such as group equivariance (Cohen and Welling, 2016a,b), $k$-Lipschitz continuity, or boundary conditions.

The input and output discretizations of the layer can be chosen independently, allowing for padding or striding (see Appendix D). The input discretization fully determines which points on $S$ are evaluated.

### 3.1. Training DI-Nets

As with discrete networks, the pipeline for training an DI-Net varies with the task. We outline steps for training a classifier and dense prediction model in Algorithms E.2 and F.3. When training an NF classifier, the discretization may be specified manually or sampled from a low discrepancy sequence to perform QMC integration. When training networks for dense prediction, the output discretization should be chosen to match the coordinates of the ground truth labels. The input discretization can be arbitrarily chosen, but in our experiments we set it equal to the output discretization. At inference time, the network can be evaluated under any desired output discretization, making the output in effect an NF. Similar pipelines can be used to train DI-Net for other tasks such as scene editing, inverse problems, or representation learning.

## 4. Experiments

We analyze our model's performance as a classifier of NFs fit to natural images (ImageNet1k) under different discretizations. We train DI-Nets with 2 and 4 MLP convolutional layers (DI-Net-2 and DI-Net-4), as well as CNNs with equivalent architectures. Appendix E provides experimental details and Appendix F provides an additional experiment on using DI-Net to segment NFs.

DI-Nets somewhat underperform their CNN counterparts (Table E.2), and the gap in performance is larger for the deeper models. However, our discretization invariant model better generalizes to higher-resolution images, where it outperforms CNNs on images at twice the resolution and has a very small drop in performance compared to the resolution it was trained on.

Having found that discretization invariance enables generalization to a different resolution at test time, we examine to what extent the network can adapt to an entirely different type of discretization at test time. We use grid, QMC (randomized Sobol sequence) and shrunk[6] discretizations of 1024 ($32 \times 32$) points.

Interestingly, changing discretization type at inference time has varying impact, usually only slightly degrading the performance of DI-Net (Table 1), but resulting in a dramatic drop in performance when shifting from a high discrep-

Table 1: DI-Net-2 accuracy under discretizations of 1024 points.

| Train→Test Type | Accuracy |
|---|---|
| QMC→Grid | 53.7% |
| Grid→QMC | 54.4% |
| QMC→Shrunk | 51.3% |
| Shrunk→QMC | 33.9% |
| Grid→Grid | 55.8% |
| QMC→QMC | 56.6% |
| Shrunk→Shrunk | **57.0%** |

ancy discretization to a low discrepancy discretization (shrunk→QMC). This suggests that discretization invariance only provides a weak guarantee on the stability of a model's behavior in the general case, and points to the importance of training on the right discretizations to attain a network that generalizes well to other discretizations.

---

6. The Shrunk discretization shrinks a Sobol sequence towards the center of the image (each point $x \in [-1, 1]^2$ is mapped to $x^2 \text{sgn}(x)$). In image classification, the object of interest is often centered in the image, which may explain why the shrunk→shrunk discretization outperforms QMC→QMC.

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

# Appendix

Appendix A describes related work. Appendix B provides additional background on quasi-Monte Carlo integration and low discrepancy sequences, including details about the conditions under which our theoretical results hold. Appendix C provides proofs of Theorems 22 and 8, as well as extensions of these theorems from the single-channel case stated in the main text to multi-channel maps. Appendix D provides a detailed specification of DI-Net layers that enable DI-Nets to replicate the behavior of CNNs. Appendix E provides additional details of the classification experiment. Appendix F provides an additional experiment on NF segmentation. Appendix G provides several analyses including properties of DI-Nets under different discretizations. Appendix H describes limitations and directions for future work.

## Appendix A.  Related Work

**Neural fields**  Multilayer perceptrons (MLPs) can be trained to capture a wide range of continuous data with high fidelity. The most prominent domains include shapes (Park et al., 2019; Mescheder et al., 2018), objects (Niemeyer et al., 2020; Müller et al., 2022), and 3D scenes (Mildenhall et al., 2020; Sitzmann et al., 2021), but previous works also apply NFs to gigapixel images (Martel et al., 2021), volumetric medical images (Corona-Figueroa et al., 2022), acoustic data (Sitzmann et al., 2020b; Gao et al., 2021), tactile data (Gao et al., 2022), depth and segmentation maps (Kundu et al., 2022), and 3D motion (Niemeyer et al., 2019). Hypernetworks and modulation networks were developed for learning directly with NFs, and have been demonstrated on tasks including generative modeling, data imputation, novel view synthesis and classification (Sitzmann et al., 2020b, 2021; Tancik et al., 2020a; Sitzmann et al., 2019, 2020a; Mehta et al., 2021; Chan et al., 2021; Dupont et al., 2021, 2022). Hypernetworks use meta-learning to learn to produce the MLP weights of desired output NFs, while modulation networks predict modulations that can be used to transform the parameters of an existing NF or generate a new NF. Another approach for learning maps between NFs in 3D space evaluates an input NF at grid points, produces features at the same points via a U-Net, and passes "raytraced" features through an MLP to produce output values from arbitrary camera angles (Vora et al., 2021).

**Discretization invariant networks**  Networks that are agnostic to the discretization of the data domain has been explored in several contexts. Hilbert space PCA, DeepONets and neural operators learn discretization invariant maps between function spaces (Bhattacharya et al., 2020; Lu et al., 2021; Li et al., 2020; Kovachki et al., 2021b), and are tailored to solve partial differential equations efficiently. On surface meshes, DiffusionNet (Sharp et al., 2022) uses the diffusion operator to achieve convergent behavior under mesh refinement. At the core of many other discretization invariant approaches is the continuous convolution, which also provides permutation invariance, translation invariance and locality. Its applications include modeling point clouds (Wang et al., 2021; Boulch, 2019), graphs (Fey et al., 2017), fluids (Ummenhofer et al., 2019), and sequential data (Romero et al., 2021b,a). These previous works define discretization invariance as convergent behavior in the limit of infinite sample points, but do not characterize how different discretizations yield different behaviors in the finite case. In this work, we choose a stronger definition that bounds the difference between any two finite discretizations, which we show implies the convergence condition.

We formulate discretization invariant networks on general metric spaces, which generalizes DeepONets and neural operators, then we focus on the continuous convolution as a core layer for vision applications.

**Approximation capabilities of neural networks**   A fundamental result in approximation theory is that the set of single-layer neural networks is dense in a large space of functionals including $L^p(\mathbb{R}^n)$ (Hornik, 1991). Subsequent works designed constructive examples using various non-linear activations (Chen et al., 1995; Chen and Chen, 1993). While this result is readily extended to multi-dimensional outputs, existing approximation results for the case of infinite dimensional outputs (e.g., $L^p(\mathbb{R}^n) \to L^p(\mathbb{R}^n)$) do not explicitly characterize the contribution of data discretization to the approximation error (Bhattacharya et al., 2020; Lanthaler et al., 2022; Kovachki et al., 2021b,a).

## Appendix B. Koksma–Hlawka inequality and low discrepancy sequences

Recall that a function $f \in L^2(\Omega)$ satisfies a Koksma–Hlawka inequality if for any point set $X \subset \Omega$,

$$\left| \frac{1}{N} \sum_{j=1}^{N} f(x') - \int_{\Omega} f(x)\,dx \right| \le V(f)\,D(X), \tag{6}$$

for normalized measure $dx$, some notion of variation $V$ of the function and some notion of discrepancy $D$ of the point set. The classical inequality gives a tight error bound for functions of bounded variation in the sense of Hardy-Krause (BVHK), a generalization of bounded variation to multivariate functions on $[0,1]^d$ which has bounded variation in each variable. Specifically, the variation is defined as:

$$V_{HK}(f) = \sum_{\alpha \in \{0,1\}^d} \int_{[0,1]^{|\alpha|}} \left| \frac{\partial^{\alpha}}{\partial x^{\alpha}} f(x_\alpha) \right| dx, \tag{7}$$

with $\{0,1\}^d$ the multi-indices and $x_\alpha \in [0,1]^d$ such that $x_{\alpha,j} = x_j$ if $j \in \alpha$ and $x_{\alpha,j} = 1$ otherwise. The classical inequality also uses the star discrepancy of the point set $X$, given by:

$$D^*(X) = \sup_{I \in J} \left| \frac{1}{N} \sum_{j=1}^{N} 1_I(x_j) - \lambda(I) \right|, \tag{8}$$

where $J$ is the set of $d$-dimensional intervals that include the origin, and $\lambda$ the Lebesgue measure.

A point set is called low discrepancy if its discrepancy is on the order of $O((\ln N)^d/N)$. Quasi-Monte Carlo calculates the sample mean using a low discrepancy sequence (see Fig. B.1 for examples in 2D), as opposed to the i.i.d. point set generated by standard Monte Carlo, which will generally be high discrepancy. Because the Koksma–Hlawka inequality is sharp, when estimating the integral of a BVHK function on $[0,1]^d$, the error of the QMC approximation decays as $O((\ln N)^d/N)$, in contrast to the error of the standard Monte Carlo approximation that decays as $O(N^{-1/2})$ (Caflisch, 1998).

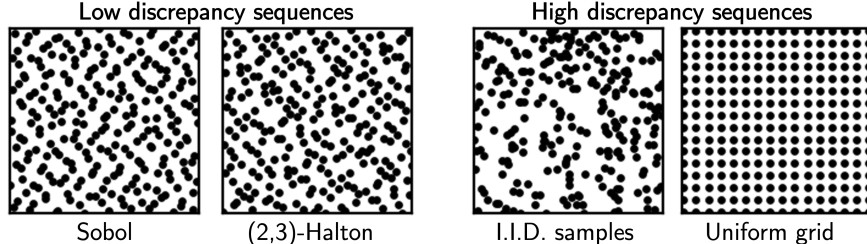

Figure B.1: Examples of low and high discrepancy sequences in 2D.

However, BVHK is a rather restrictive class of functions defined on $[0,1]^d$ that excludes all functions with discontinuities. Brandolini et al. (2013) extended the Koksma–Hlawka inequality to two classes of functions defined below:

**Piecewise smooth functions** Let $f$ be a smooth function on $[0,1]^d$ and $\Omega$ a Borel subset of $[0,1]^d$. Then $f|_\Omega$ is a piecewise smooth function with the Koksma–Hlawka inequality given by variation

$$V(f) = \sum_{\alpha \in \{0,1\}^d} 2^{d-|\alpha|} \int_{[0,1]^d} \left| \frac{\partial^\alpha}{\partial x^\alpha} f(x) \right| \, dx, \tag{9}$$

and discrepancy

$$D(X) = 2^d \sup_{I \subseteq [0,1]^d} \left| \frac{1}{N} \sum_{j=1}^{N} 1_{\Omega \cap I}(x_j) - \lambda(\Omega \cap I) \right|. \tag{10}$$

$W^{d,1}$ **functions on manifolds** Let $M$ be a smooth compact $d$-dimensional manifold with normalized measure $dx$. Given local charts $\{\phi_k\}_{k=1}^{K}$, $\phi_k : [0,1]^d \to M$, the variation of a function $f \in W^{d,1}(M)$ is characterized as:

$$V(f) = c \sum_{k=1}^{K} \sum_{|\alpha| \leq n} \int_{[0,1]^d} \left| \frac{\partial^\alpha}{\partial x^\alpha} (\psi_k(\phi_k(x)) f(\phi_k(x)) \right| \, dx, \tag{11}$$

with $\{\psi_k\}_{k=1}^{K}$ a smooth partition of unity subordinate to the charts, and constant $c > 0$ that depends on the charts but not on $f$. Defining the set of intervals in $M$ as $J = \{U : U = \phi_k(I)$ for some $k$ and $I \subseteq [0,1]^d\}$, with measure $\mu(U) = \lambda(I)$, the discrepancy of a point set $Y = \{y_j\}_{y=1}^{N}$ on $M$ is:

$$D(Y) = \sup_{U \in J} \left| \frac{1}{N} \sum_{j=1}^{N} 1_U(y_j) - \mu(U) \right|. \tag{12}$$

The notion of discrepancy is not limited to the Lebesgue measure. The existence of low discrepancy point sets has been proven for non-negative, normalized Borel measures on $[0,1]^d$ due to Aistleitner and Dick (2013). An extension of our framework to non-uniform measures is a promising direction for future work (see Section H).

## Appendix C. Proofs

### C.1. Proof of Property 8 (Universal Approximator)

**Definition 9** *Let $\mathcal{G}, \mathcal{G}'$ be subsets of $L^2(\Omega)$ restricted to compactly supported functions of absolute bounded variation. By this we mean there exists $V^*$ such that every $f \in \mathcal{G} \cup \mathcal{G}'$ satisfies a Koksma–Hlawka inequality (6) with $V(|f|) < V^*$.*

$\mathcal{G}, \mathcal{G}'$ are bounded in $L^1$ norm since all their functions are compactly supported and bounded.

Consider a Lipschitz continuous map $\mathcal{R} : \mathcal{G} \to \mathcal{G}'$ such that $d(\mathcal{R}[f], \mathcal{R}[g])_{L^1} \leq M_0 d(f, g)_{L^1}$ for some constant $M_0$ and all $f, g \in \mathcal{G}$. Let $M = \max\{M_0, 1\}$.

Select a fixed point set $X = \{x_j\}_{j=1}^N$ in $\Omega$ with discrepancy $D(X) = \frac{\epsilon}{12(M+2)V^*}$. By (6) this yields:

$$\left| \frac{1}{N} \sum_{j=1}^N f(x_j) - \int_\Omega f(x)\, dx \right| \leq \frac{\epsilon}{12(M+2)}, \tag{13}$$

for all $f \in \mathcal{G} \cup \mathcal{G}'$.

**Definition 10** *The projection $\pi : f \mapsto \mathbf{f}$ is a quotient map $L^2(\Omega) \to L^2(\Omega)/\sim$ under the equivalence relation $f \sim g$ iff $f(x_j) = g(x_j)$ for all $x_j \in X$.*

$L^2(\Omega)/\sim$ can be identified with $\mathbb{R}^N$, and thus can be given the normalized $\ell^1$ norm:

$$\|\pi f\|_{\ell^1} = \frac{1}{N} \sum_{j=1}^N |f(x_j)|. \tag{14}$$

**Definition 11** *Denote the preimage of $\pi$ as $\pi^{-1} : \mathbf{f}' \mapsto \{f' \in \mathcal{G}' : \pi f' = \mathbf{f}'\}$. Invoking the axiom of choice, define the inverse projection $\pi^{-1} : \pi\mathcal{G}' \to \mathcal{G}'$ by a choice function over the sets $\pi^{-1}(\pi\mathcal{G}')$.*

**Remark:** Note that this inverse projection corresponds to some way of interpolating the $N$ sample points such that the output is in $\mathcal{G}'$. Although our definition implies the existence of such an interpolator, we leave its specification as an open problem. Since $\Omega$ only permits discontinuities along a fixed Borel subset of $[0, 1]^d$, these boundaries can be specified *a priori* in the interpolator. Since all functions in $\mathcal{G}'$ are bounded and continuous outside this set, the interpolator can be represented by a bounded continuous map, hence it is expressible by a DI-Net layer.

**Definition 12** *$\pi$ generates a $\sigma$-algebra on $\mathcal{G}$ given by $A = \{\pi^{-1}(S) : S \in L\}$, with $L$ the $\sigma$-algebra of Lebesgue measurable sets on $\mathbb{R}^N$. Because this $\sigma$-algebra depends on $\epsilon$ and the Lipschitz constant of $\mathcal{R}$ via the point set discrepancy, we may write it as $A_{\epsilon, \mathcal{R}}$.*

**Remark:** In this formulation, we let the tolerance $\epsilon$ and the Lipschitz constant of $\mathcal{R}$ dictate what subsets of $\mathcal{G}$ are measurable, and thus which measures on $\mathcal{G}$ are permitted. However, if the desired measure $\nu$ is more fine-grained than what is permitted by $A_{\epsilon,\mathcal{R}}$, then it is $\nu$ that should determine the number of QMC samples $N$, rather than $\epsilon$ or $\mathcal{R}$.

We now state the following lemmas which will be used to prove our universal approximation theorem.

**Lemma 13** *There is a map $\tilde{\mathcal{R}} : \pi\mathcal{G} \to \pi\mathcal{G}'$ such that*

$$\int_{\Omega} \left| \mathcal{R}[f](x) - \pi^{-1} \circ \tilde{\mathcal{R}} \circ \pi[f](x) \right| dx = \frac{\epsilon}{6}. \tag{15}$$

**Proof** Let $g(x) = |f(x)|$ for $f \in \mathcal{G} \cup \mathcal{G}'$. Because (13) applies to $g(x)$, we have:

$$\left| \frac{1}{N} \sum_{j=1}^{N} g(x_j) - \int_{\Omega} g(x)\, dx \right| \leq \frac{\epsilon}{12(M+2)} \tag{16}$$

$$\left| \|\pi f\|_{\ell^1} - \|f\|_{L^1} \right| \leq \frac{\epsilon}{12(M+2)}. \tag{17}$$

Eqn. (17) also implies that for any $\mathbf{f} \in \pi\mathcal{G} \cup \pi\mathcal{G}'$, we have:

$$\left| \|\mathbf{f}\|_{\ell^1} - \|\pi^{-1}\mathbf{f}\|_{L^1} \right| \leq \frac{\epsilon}{12(M+2)}. \tag{18}$$

Combining (17) and (18), we obtain

$$\left| \|f\|_{L^1} - \|\pi^{-1} \circ \pi[f]\|_{L^1} \right| \leq \frac{\epsilon}{6(M+2)}. \tag{19}$$

By the triangle inequality and applying $\mathcal{R}$:

$$\int_{\Omega} \left| \mathcal{R}[f](x) - \pi^{-1} \circ \pi \circ \mathcal{R}[f](x) \right| dx \leq \frac{\epsilon}{6(M+2)}. \tag{20}$$

For any $f, g \in \mathcal{G}$ such that $\pi f = \pi g$, (17) tells us that $d(f,g)_{L^1}$ is at most $\epsilon/6(M+2)$. Recall $M$ was defined such that $d(\mathcal{R}[f], \mathcal{R}[g])_{L^1} \leq M d(f,g)_{L^1}$ for any $\mathcal{R}$.

$$d(\pi \circ \mathcal{R}[f], \pi \circ \mathcal{R}[g])_{L^1} \leq \frac{M\epsilon}{6(M+2)} + \frac{\epsilon}{6(M+2)} \tag{21}$$

$$= \frac{(M+1)}{(M+2)} \frac{\epsilon}{6} \tag{22}$$

So defining:

$$\tilde{\mathcal{R}} = \arg\min_{H} d(H \circ \pi[f], \pi \circ \mathcal{R}[f])_{\ell^1}, \tag{23}$$

we have

$$\left| \tilde{\mathcal{R}} \circ \pi[f] - \pi \circ \mathcal{R}[f] \right| \leq \frac{(M+1)}{(M+2)} \frac{\epsilon}{6}. \tag{24}$$

Then by (20),

$$\int_\Omega \left| \mathcal{R}[f](x) - \pi^{-1} \circ \tilde{\mathcal{R}} \circ \pi[f](x) \right| dx \leq \frac{\epsilon}{6(M+2)} + \frac{(M+1)}{(M+2)} \frac{\epsilon}{6} \tag{25}$$

$$= \frac{\epsilon}{6}. \tag{26}$$

∎

**Lemma 14** *Consider the extension of $\tilde{\mathcal{R}}$ to $\mathbb{R}^N \to \mathbb{R}^N$ in which each component of the output has the form:*

$$\tilde{\mathcal{R}}_j(\mathbf{f}) = \begin{cases} \mathcal{R}[\pi^{-1}\mathbf{f}](x_j) & \text{if } \mathbf{f} \in \pi\mathcal{G} \\ 0 & \text{otherwise.} \end{cases} \tag{27}$$

*Then any finite measure $\nu$ on the measurable space $(\mathcal{G}, A)$ induces a finite measure $\mu$ on $(\mathbb{R}^N, L)$, and $\int_{\mathbb{R}^N} |\tilde{\mathcal{R}}_j(\mathbf{f})| \mu(d\mathbf{f}) < \infty$ for each $j$.*

**Proof** Since the $\sigma$-algebra $A$ on $\mathcal{G}$ is generated by $\pi$, the measure $\mu : \mu(\pi S) = \nu(S)$ for all $S \in A$ is finite and defined w.r.t. the Lebesgue measurable sets on $\pi\mathcal{G}$. Since $\pi\mathcal{G}$ can be identified with a measurable subset of $\mathbb{R}^N$, $\mu$ can be naturally extended to $\mathbb{R}^N$. Doing so makes it absolutely continuous w.r.t. the Lebesgue measure on $\mathbb{R}^N$.

To show $\tilde{\mathcal{R}}_j(\mathbf{f})$ is integrable, it is sufficient to show it is bounded and compactly supported.

$\mathcal{G}$ is bounded in the $L^1$ norm. Thus by (17), $\pi\mathcal{G}$ is bounded in the normalized $\ell_1$ norm. The $\ell_1$ norm in $\mathbb{R}^N$ is strongly equivalent to the uniform norm, so there is some compact set $[-c, c]^N$, $c > 0$ for which the extension of $\pi\mathcal{G}$ to $\mathbb{R}^N$ vanishes, so $\text{supp}(\tilde{\mathcal{R}}_j(\mathbf{f})) \subseteq [-c, c]^N$.

Similarly, $\pi\mathcal{G}'$ is bounded in the $\ell^1$ norm, hence there exists $c'$ such that $\tilde{\mathcal{R}}_j < c'$ for all $j$. ∎

**Lemma 15** *For any finite measure $\mu$ absolutely continuous w.r.t. the Lebesgue measure on $\mathbb{R}^n$, $J \in L^1(\mu)$ and $\epsilon > 0$, there is a network $\mathcal{K}$ such that:*

$$\int_{\mathbb{R}^n} |J(\mathbf{f}) - \mathcal{K}(\mathbf{f})| \, \mu(d\mathbf{f}) < \frac{\epsilon}{2}. \tag{28}$$

**Proof** The following construction is adapted from lu2017expressive. Since $J$ is integrable, there is a cube $E = [-c, c]^n$ such that:

$$\int_{\mathbb{R}^n \setminus E} |J(\mathbf{f})| \mu(d\mathbf{f}) < \frac{\epsilon}{8} \tag{29}$$

$$\|J - 1_E J\|_1 < \frac{\epsilon}{8}. \tag{30}$$

*Case 1: $J$ is non-negative on all of $\mathbb{R}^n$*
Define the set under the graph of $J|_E$:

$$G_{E,J} \triangleq \{(\mathbf{f}, y) : \mathbf{f} \in E, y \in [0, J(\mathbf{f})]\}. \tag{31}$$

$G_{E,J}$ is compact in $\mathbb{R}^{n+1}$, hence there is a finite cover of open rectangles $\{R_i'\}$ satisfying $\mu(\cup_i R_i') - \mu(G_{E,J}) < \frac{\epsilon}{8}$ on $\mathbb{R}^n$. Take their closures, and extend the sides of all rectangles indefinitely. This results in a set of pairwise almost disjoint rectangles $\{R_i\}$. Taking only the rectangles $R = \{R_i : \mu(R_i \cap G_{E,J}) > 0\}$ results in a finite cover satisfying:

$$\sum_{i=1}^{|R|} \mu(R_i) - \mu(G_{E,J}) < \frac{\epsilon}{8}. \tag{32}$$

This implies:

$$\sum_{i=1}^{|R|} \mu(R_i) < \|J\|_1 + \frac{\epsilon}{8}, \tag{33}$$

and also,

$$\frac{\epsilon}{8} > \sum_{i=1}^{|R|} \int_{\mathbb{R}^n} 1_{R_i}(\mathbf{f}, J(\mathbf{f}))\, \mu(d\mathbf{f}) + \|J\|_1 \tag{34}$$

$$\geq \int_E |J(\mathbf{f}) - \sum_{i=1}^{|R|} 1_{R_i}(\mathbf{f}, J(\mathbf{f}))|\, \mu(d\mathbf{f}), \tag{35}$$

by the triangle inequality. For each $R_i = [a_{i1}, b_{i1}] \times \ldots [a_{in}, b_{in}] \times [\zeta_i, \zeta_i + y_i]$, let $X_i$ be its first $n$ components (i.e., the projection of $R_i$ onto $\mathbb{R}^n$). Then we have

$$\int_E |J(\mathbf{f}) - \sum_{i=1}^{|R|} y_i 1_{X_i}(\mathbf{f})|\, \mu(d\mathbf{f}) < \frac{\epsilon}{8}. \tag{36}$$

Let $Y(\mathbf{f}) \triangleq \sum_{i=1}^{|R|} y_i 1_{X_i}(\mathbf{f})$. By the triangle inequality,

$$\int_{\mathbb{R}^n} |J(\mathbf{f}) - \mathcal{K}(\mathbf{f})|\, \mu(d\mathbf{f}) \leq \|J - 1_E J\|_1 + \|1_E J - Y\|_1 + \|\mathcal{K} - Y\|_1 \tag{37}$$

$$< \frac{\epsilon}{4} + \|\mathcal{K} - Y\|_1, \tag{38}$$

by (30) and (36). So it remains to construct $\mathcal{K}$ such that $\|\mathcal{K} - Y\|_1 < \frac{\epsilon}{4}$. Because $1_{X_i}$ is discontinuous at the boundary of the rectangle $X_i$, it cannot be produced directly from a DI-Net (recall that all layers are continuous maps). However, we can approximate it arbitrarily well with a piece-wise linear function that rapidly ramps from 0 to 1 at the boundary.

For fixed rectangle $X_i$ and $\delta \in (0, 0.5)$, consider the inner rectangle $X_\delta \subset X_i$:

$$X_\delta = (a_1 + \delta(b_1 - a_1), b_1 - \delta(b_1 - a_1)) \times \cdots \times (a_n + \delta(b_n - a_n), b_n - \delta(b_n - a_n)), \tag{39}$$

where we omit subscript $j$ for clarity. Letting $b_i' = b_i - \delta(b_i - a_i)$, define the function:

$$T(\mathbf{f}) = \prod_{i=1}^n \frac{1}{\delta} \Big[ \texttt{ReLU}(\delta - \texttt{ReLU}(\mathbf{f}_i - b_i')) - \texttt{ReLU}(\delta - \texttt{ReLU}(\mathbf{f}_i - a_i)) \Big], \tag{40}$$

where $\texttt{ReLU}(x) = \max(x, 0)$. $T(\mathbf{f})$ is a piece-wise linear function that ramps from 0 at the boundary of $X_i$ to 1 within $X_\delta$, and vanishes outside $X_i$. Note that

$$\|1_X - T\|_1 < \mu(X) - \mu(X_\delta) \tag{41}$$

$$= (1 - (1 - 2\delta)^n)\mu(X), \tag{42}$$

if $\mu$ is the Lebesgue measure. $\delta$ may need to be smaller under other measures, but this adjustment is independent of the input $\mathbf{f}$ so it can be specified *a priori*.

Recall that the function we want to approximate is $Y(\mathbf{f}) = \sum_{i=1}^{|R|} y_i 1_{X_i}(\mathbf{f})$. We can build NF-Net layers $\mathcal{K} : \mathbf{f} \mapsto \mathcal{K}(\mathbf{f}) = \sum_{i=1}^{|R|} y_i T_i(\mathbf{f})$, since this only involves linear combinations and ReLUs. Then,

$$\|\mathcal{K} - Y\|_1 = \int_{\mathbb{R}^n} \sum_{i=1}^{|R|} y_i \left(T_i(\mathbf{f}) - 1_{X_i}(\mathbf{f})\right) d\mathbf{f} \tag{43}$$

$$= \sum_{i=1}^{|R|} y_i \|1_{X_i} - T_i\|_1 \tag{44}$$

$$< (1 - (1 - 2\delta)^n) \sum_{i=1}^{|R|} y_i \mu(X_i) \tag{45}$$

$$= (1 - (1 - 2\delta)^n) \sum_{i=1}^{|R|} \mu(R_i) \tag{46}$$

$$< (1 - (1 - 2\delta)^n) \left(\|J\|_1 + \frac{\epsilon}{8}\right), \tag{47}$$

by (33). And so by choosing:

$$\delta = \frac{1}{2} \left(1 - \left(1 - \frac{\epsilon}{4}\left(\|J\|_1 + \frac{\epsilon}{8}\right)^{-1}\right)^{1/n}\right), \tag{48}$$

we have our desired bound $\|\mathcal{K} - Y\|_1 < \frac{\epsilon}{4}$ and thereby $\|J - \mathcal{K}\|_1 < \frac{\epsilon}{2}$.
*Case 2: $J$ is negative on some region of $\mathbb{R}^n$*
Letting $J^+(\mathbf{f}) = \max(0, J(\mathbf{f}))$ and $J^-(\mathbf{f}) = \max(0, -J(\mathbf{f}))$, define:

$$G_{E,J}^+ \triangleq \{(\mathbf{f}, y) : \mathbf{f} \in E, y \in [0, J^+(\mathbf{f})]\} \tag{49}$$

$$G_{E,J}^- \triangleq \{(\mathbf{f}, y) : \mathbf{f} \in E, y \in [0, J^-(\mathbf{f})]\}. \tag{50}$$

As in (32), construct covers of rectangles $R^+$ over $G_{E,J}^+$ and $R^-$ over $G_{E,J}^-$ each with bound $\frac{\epsilon}{16}$ and $\mathbb{R}^n$ projections $X^+$, $X^-$. Let:

$$Y^+(\mathbf{f}) = \sum_{i=1}^{|R^+|} y_i^+ 1_{X_i^+}(\mathbf{f}) \tag{51}$$

$$Y^-(\mathbf{f}) = \sum_{i=1}^{|R^-|} y_i^- 1_{X_i^-}(\mathbf{f}) \tag{52}$$

$$Y = Y^+ - Y^- \tag{53}$$

We can derive an equivalent expression to (36):

$$\frac{\epsilon}{8} > \int_E \left| J(\mathbf{f}) - \sum_{i=1}^{|R^+|} y_i^+ 1_{X_i^+}(\mathbf{f}) + \sum_{i=1}^{|R^-|} y_i^- 1_{X_i^-}(\mathbf{f}) \right| d\mathbf{f} \tag{54}$$

$$= \|1_E J - Y\|_1. \tag{55}$$

Similarly to earlier, we use (30) and (55) to get:

$$\int_{\mathbb{R}^n} |J(\mathbf{f}) - \mathcal{K}(\mathbf{f})| \, d\mathbf{f} < \frac{\epsilon}{4} + \|\mathcal{K} - Y\|_1. \tag{56}$$

Choosing $T_i^+(\mathbf{f})$ and $T_i^-(\mathbf{f})$ the piece-wise linear functions associated with $X_i^+$ and $X_i^-$, and:

$$\mathcal{K}(\mathbf{f}) = \sum_{i=1}^{|R^+|} y_i^+ T_i^+(\mathbf{f}) - \sum_{i=1}^{|R^-|} y_i^- T_i^-(\mathbf{f}), \tag{57}$$

we have:

$$\|\mathcal{K} - Y\|_1 = \int_{\mathbb{R}^n} \left| \sum_{i=1}^{|R^+|} y_i^+ \left( T_i^+(\mathbf{f}) - 1_{X_i^+}(\mathbf{f}) \right) - \sum_{i=1}^{|R^-|} y_i^- \left( T_i^-(\mathbf{f}) - 1_{X_i^-}(\mathbf{f}) \right) \right| d\mathbf{f}, \tag{58}$$

applying the triangle inequality,

$$\leq \sum_{i=1}^{|R^+|} y_i^+ \left\| 1_{X_i^+} - T_i^+ \right\|_1 + \sum_{i=1}^{|R^-|} y_i^- \left\| 1_{X_i^-} - T_i^- \right\|_1 \tag{59}$$

$$< (1 - (1 - 2\delta^+)^n) \sum_{i=1}^{|R^+|} y_i^+ \mu(X_i^+) + (1 - (1 - 2\delta^-)^n) \sum_{i=1}^{|R^-|} y_i^- \mu(X_i^-) \tag{60}$$

$$< (1 - (1 - 2\delta^+)^n) \left( \|J^+\|_1 + \frac{\epsilon}{16} \right) + (1 - (1 - 2\delta^-)^n) \left( \|J^-\|_1 + \frac{\epsilon}{16} \right). \tag{61}$$

By choosing:

$$\delta^+ = \frac{1}{2} \left( 1 - \left( 1 - \frac{\epsilon}{8} \left( \|J^+\|_1 + \frac{\epsilon}{16} \right)^{-1} \right)^{1/n} \right) \tag{62}$$

$$\delta^- = \frac{1}{2} \left( 1 - \left( 1 - \frac{\epsilon}{8} \left( \|J^-\|_1 + \frac{\epsilon}{16} \right)^{-1} \right)^{1/n} \right), \tag{63}$$

and proceeding as before, we arrive at the same bounds $\|\mathcal{K} - Y\|_1 < \frac{\epsilon}{4}$ and $\|J - \mathcal{K}\|_1 < \frac{\epsilon}{2}$.

Putting it all together, Algorithm C.1 implements the network logic for producing the function $\mathcal{K}$.

---

**Algorithm C.1** DI-Net approximation of $\mathbf{f} \mapsto J(\mathbf{f})$

---

*Setup* **Input:** target function $J$, $L_1$ tolerance $\epsilon/2$

Choose rectangles $R_i^+ = [a_{i1}^+, b_{i1}^+] \times \ldots [a_{in}^+, b_{in}^+] \times [\zeta_i^+, \zeta_i^+ + y_i^+]$ satisfying (32) and $R^-$ similarly

$\delta^+ \leftarrow \frac{1}{2} \left( 1 - (1 - \frac{\epsilon}{8} \left( \|J^+\|_1 + \frac{\epsilon}{16} \right)^{-1})^{1/n} \right)$   $\delta^- \leftarrow \frac{1}{2} \left( 1 - (1 - \frac{\epsilon}{8} \left( \|J^-\|_1 + \frac{\epsilon}{16} \right)^{-1})^{1/n} \right)$

*Inference* **Input:** discretized input $\mathbf{f} = \{\mathbf{f}_k\}_{k=1}^n$

$x \leftarrow (0, 0, 1, 0, 0)$ **for** rectangle $R_i^+ \in R^+$ **do**

> **for** dimension $k \in 1 : n$ **do**
>
> > $x \leftarrow (\mathbf{f}_k - b_{ik}^+ + \delta(b_{ik}^+ - a_{ik}^+), \mathbf{f}_k - a_{ik}^+, x_3, x_4, x_5)$   $x \leftarrow \texttt{ReLU}(x)$   $x \leftarrow (\delta - x_1, \delta - x_2, x_3, x_4, x_5)$   $x \leftarrow \texttt{ReLU}(x)$   $x \leftarrow (0, 0, x_3(x_1 - x_2)/\delta, x_4, x_5)$
>
> **end**
>
> $x \leftarrow (0, 0, 1, y_i^+ x_3 + x_4, x_5)$

**end**

**for** rectangle $R_i^- \in R^-$ **do**

> **for** dimension $k \in 1 : n$ **do**
>
> > $x \leftarrow (\mathbf{f}_k - b_{ik}^- + \delta(b_{ik}^- - a_{ik}^-), \mathbf{f}_k - a_{ik}^-, x_3, x_4, x_5)$   $\ldots$
>
> **end**
>
> $x \leftarrow (0, 0, 1, x_4, y_i^- x_3 + x_5)$

**end**

**Output:** $x_4 - x_5$

---

We can provide $x$ with access to $\mathbf{f}$ either through skip connections or by appending channels with the values $\{c + \mathbf{f}_k\}_{k=1}^n$ (which will be preserved under ReLU). ∎

**Theorem 16 (Maps between Single-Channel NFs)** *For any Lipschitz continuous map* $\mathcal{R} : \mathcal{G} \to \mathcal{G}'$, *any* $\epsilon > 0$, *and any finite measure* $\nu$ *w.r.t. the measurable space* $(\mathcal{G}, A_{\epsilon, \mathcal{R}})$, *there exists a DI-Net* $\mathcal{T}$ *that satisfies:*

$$\int_{\mathcal{G}} \|\mathcal{R}(f) - \mathcal{T}(f)\|_{L^1(\Omega)} \nu(df) < \epsilon. \tag{64}$$

**Proof** If $\nu$ is not normalized, the discrepancy of our point set needs to be further divided by $\max\{\nu(\mathcal{G}), 1\}$. We assume for the remainder of this section that $\nu$ is normalized. Perform the construction of Lemma 15 $N$ times, each with a tolerance of $\epsilon/2N$. Choose a partition of unity $\{\psi_j\}$ for which $\psi_j(x_k) = \delta_{jk}$, and output $N$ channels with the values $\{\mathcal{K}_j(\mathbf{f})\psi_j(\cdot)\}_{j=1}^N$. By summing these channels we obtain a network $\tilde{\mathcal{K}}$ that fully specifies the desired behavior of $\tilde{\mathcal{R}} : \mathbb{R}^N \to \mathbb{R}^N$, with combined error:

$$\int_{\mathbb{R}^N} \left\| \tilde{\mathcal{R}}(\mathbf{f}) - \tilde{\mathcal{K}}(\mathbf{f}) \right\|_{\ell^1} \mu(d\mathbf{f}) < \frac{\epsilon}{2}. \tag{65}$$

Thus,

$$\int_{\mathcal{G}} \left| \frac{1}{N} \sum_{j=1}^N \tilde{\mathcal{R}} \circ \pi[f](x_j) - \tilde{\mathcal{K}} \circ \pi[f](x_j) \right| \nu(df) \leq \frac{\epsilon}{2}. \tag{66}$$

By (18) we have:

$$\int_{\mathcal{G}} \left| \int_{\Omega} \left| \pi^{-1} \circ \tilde{\mathcal{R}} \circ \pi[f](x) - \pi^{-1} \circ \tilde{\mathcal{K}} \circ \pi[f](x) \right| dx \right| \nu(df) \leq \frac{\epsilon}{2} + \frac{\epsilon}{6(M+2)} \tag{67}$$

By Lemma 13 we have:

$$\int_{\mathcal{G}} \int_{\Omega} \left| \mathcal{R}[f](x) - \pi^{-1} \circ \tilde{\mathcal{K}} \circ \pi[f](x) \right| dx \, \nu(df) \leq \frac{\epsilon}{2} + \frac{\epsilon}{6(M+2)} + \frac{\epsilon}{6} \tag{68}$$

And thus the network $\mathcal{T} = \pi^{-1} \circ \tilde{\mathcal{K}} \circ \pi$ gives us the desired bound:

$$\int_{\mathcal{G}} \|\mathcal{R}(f) - \mathcal{T}(f)\|_{L^1(\Omega)} \, \nu(df) < \epsilon. \tag{69}$$

∎

**Corollary 17 (Maps from NFs to vectors)** *For any Lipschitz continuous map $\mathcal{R} : \mathcal{G} \to \mathbb{R}^n$, any $\epsilon > 0$, and any finite measure $\nu$ w.r.t. the measurable space $(\mathcal{G}, A_{\epsilon,\mathcal{R}})$, there exists a DI-Net $\mathcal{T}$ that satisfies:*

$$\int_{\mathcal{G}} \|\mathcal{R}(f) - \mathcal{T}(f)\|_{\ell_1(\mathbb{R}^n)} \nu(df) < \epsilon. \tag{70}$$

**Proof** Let $M_0$ be the Lipschitz constant of $\mathcal{R}$ in the sense that $d(\mathcal{R}[f], \mathcal{R}[g])_{\ell^1} \leq M_0 d(f,g)_{L^1}$. Let $M = \max\{M_0, 1\}$. There exists $\tilde{\mathcal{R}} : \pi\mathcal{G} \to \mathbb{R}^n$ such that $\left\| \tilde{\mathcal{R}} \circ \pi[f] - \mathcal{R}[f] \right\|_{\ell^1} \leq \epsilon/12$. As in Lemma 14, consider the extension of $\tilde{\mathcal{R}}$ to $\mathbb{R}^N \to \mathbb{R}^n$ in which each component of the output has the form:

$$\tilde{\mathcal{R}}_j(\mathbf{f}) = \begin{cases} \mathcal{R}[\pi^{-1}\mathbf{f}]_j & \text{if } \mathbf{f} \in \pi\mathcal{G} \\ 0 & \text{otherwise.} \end{cases} \tag{71}$$

Then for similar reasoning, $\nu$ on $\mathcal{G}$ induces a measure $\mu$ on $\mathbb{R}^N$ that is finite and absolutely continuous w.r.t. the Lebesgue measure, and $\int_{\mathbb{R}^N} |\tilde{\mathcal{R}}_j(\mathbf{f})| \mu(d\mathbf{f}) < \infty$ for each $j$.

We construct our $\mathbb{R}^N \to \mathbb{R}$ approximation $n$ times with a tolerance of $\epsilon/2n$, such that:

$$\int_{\mathbb{R}^N} \left\| \tilde{\mathcal{R}}(\mathbf{f}) - \tilde{\mathcal{K}}(\mathbf{f}) \right\|_{\ell^1(\mathbb{R}^n)} \mu(d\mathbf{f}) < \frac{\epsilon}{2}. \tag{72}$$

Applying (17), we find that the network $\mathcal{T} = \tilde{\mathcal{K}} \circ \pi$ gives us the desired bound:

$$\int_{\mathcal{G}} \|\mathcal{R}(f) - \mathcal{T}(f)\|_{\ell_1(\mathbb{R}^n)} \nu(df) < \epsilon. \tag{73}$$

∎

**Corollary 18 (Maps from vectors to NFs)** *For any Lipschitz continuous map $\mathcal{R} : \mathbb{R}^n \to \mathcal{G}$ and any $\epsilon > 0$, there exists a DI-Net $\mathcal{T}$ that satisfies:*

$$\int_{\mathbb{R}^n} \|\mathcal{R}(x) - \mathcal{T}(x)\|_{L^1(\Omega)} dx < \epsilon. \tag{74}$$

**Proof** Define the map $\tilde{\mathcal{R}} : \mathbb{R}^n \to \pi\mathcal{G} \subset \mathbb{R}^N$ by $\tilde{\mathcal{R}} = \pi \circ \mathcal{R}$. Since $\tilde{\mathcal{R}}$ is bounded and compactly supported, $\int_{\mathbb{R}^N} |\tilde{\mathcal{R}}_i(x)| dx < \infty$ for each $i$.

We construct a $\mathbb{R}^n \to \mathbb{R}$ approximation $N$ times each with a tolerance of $\epsilon/2N$, such that:

$$\int_{\mathbb{R}^n} \left\| \tilde{\mathcal{R}}(x) - \tilde{\mathcal{K}}(x) \right\|_{L^1(\Omega)} dx < \frac{\epsilon}{2}. \tag{75}$$

Applying (18), we find that the network $\mathcal{T} = \pi^{-1} \circ \tilde{\mathcal{K}}$ gives us the desired bound:

$$\int_{\mathbb{R}^n} \|\mathcal{R}(x) - \mathcal{T}(x)\|_{L^1(\Omega)} dx < \epsilon. \tag{76}$$

$\blacksquare$

Denote the space of multi-channel NFs as $\mathcal{F}_c = \{f : \Omega \to \mathbb{R}^c : \int_\Omega \|f\|_1 dx < \infty, \ f_i \in \mathcal{G}$ for each $i\}$. Denote the norm on this space as:

$$\|f\|_{\mathcal{F}_c} = \int_\Omega \sum_{i=1}^c |f_i(x)| dx. \tag{77}$$

$\mathcal{F}_1$ is identified with $\mathcal{G}$.

**Definition 19 (Concatenation)** *Concatenation is a map from two NF channels $f_i, f_j \in \mathcal{G}$ to $[f_i, f_j] \in \mathcal{F}_2$. The concatenation of NFs can be defined inductively to yield $\mathcal{F}_n \times \mathcal{F}_m \to \mathcal{F}_{n+m}$ for any $n, m \in \mathbb{N}$.*

**Remark:** All maps $\mathcal{F}_n \times \mathcal{F}_m \to \mathcal{F}_c$ can be expressed as a concatenation followed by a map $\mathcal{F}_{n+m} \to \mathcal{F}_c$. A map $\mathbb{R}^n \to \mathcal{F}_m$ is also equivalent to $m$ maps $\mathbb{R}^n \to \mathcal{G}$ followed by concatenation. Thus, we need only characterize the maps that take one multi-channel NF as input.

Considering the maps $\mathcal{F}_n \to \mathcal{F}_m$, we choose a lower discrepancy point set $X$ on $\Omega$ such that the Koksma–Hlawka inequality yields a bound of $\epsilon/12mn(M+2)$. Let $\pi$ project each component of the input to $\pi\mathcal{G}$, and $\pi^{-1}$ inverts this projection under some choice function. We take $A'$ to be the product $\sigma$-algebra generated from this $\pi$: $A' = \{E_1 \times \cdots \times E_c : E_1, \ldots, E_c \in A\}$ where $A$ is the $\sigma$-algebra on $\mathcal{G}$ from Definition 12.

**Corollary 20 (Maps between multi-channel NFs)** *For any Lipschitz continuous map $\mathcal{R} : \mathcal{F}_n \to \mathcal{F}_m$, any $\epsilon > 0$, and any finite measure $\nu$ w.r.t. the measurable space $(\mathcal{F}_n, A'_{\epsilon,\mathcal{R}})$, there exists a DI-Net $\mathcal{T}$ that satisfies:*

$$\int_{\mathcal{F}_n} \|\mathcal{R}(f) - \mathcal{T}(f)\|_{\mathcal{F}_m} \nu(df) < \epsilon. \tag{78}$$

**Proof** The proof is very similar to that of Theorem 16. Our network now requires $nN$ maps from $\mathbb{R}^{mN} \to \mathbb{R}$ each with error $\epsilon/2mnN$. Summing the errors across all input and output channels yields our desired bound. $\blacksquare$

The multi-channel analogue of Corollary 17 is clear, and we state it here for completeness:

**Corollary 21 (Maps from multi-channel NFs to vectors)** *For any Lipschitz continuous map $\mathcal{R} : \mathcal{F}_n \to \mathbb{R}^m$, any $\epsilon > 0$, and any finite measure $\nu$ w.r.t. the measurable space $(\mathcal{F}_n, A'_{\epsilon,\mathcal{R}})$, there exists a DI-Net $\mathcal{T}$ that satisfies:*

$$\int_{\mathcal{F}_n} \|\mathcal{R}(f) - \mathcal{T}(f)\|_{\ell_1(\mathbb{R}^m)} \nu(df) < \epsilon. \tag{79}$$

### C.2. Convergent Empirical Gradients

Here we establish that backpropagation is sound, by proving the convergence of derivatives through the empirical measure in the following sense:

**Theorem 22 (Convergent Empirical Gradients)** *A DI-Net permits backpropagation of its outputs with respect to its input as well as all its learnable parameters. Under a sequence of discretizations tending to 0 discrepancy, the gradients of each layer converge to the appropriate derivative under the measure on $\Omega$.*

To establish convergence in the number of QMC samples $N$, let us suppose we have a family of sequences $\{X_N\}_{N \in \mathbb{N}} : X_N = \{x_j\}_{j=1}^N$ on $\Omega$ whose discrepancy converges to 0 as $N \to \infty$.

We note that no convergence statement is needed if the layer does not perform QMC integration. This includes layers which take $\mathbb{R}^n$ as input, as well as point-wise transformations. Then the (sub)derivatives with respect to inputs and parameters need only be well-defined at each point of the output in order to enable backpropagation.

Thus we consider a layer $\mathcal{H}_\phi$ which takes an NF $f$ as input. By (3), it can be written as:

$$\mathcal{H}_\phi[f] = \int_\Omega H_\phi[f](x)dx. \tag{80}$$

We write the QMC estimate of $\mathcal{H}_\phi[f]$ under $X_N$ as:

$$\hat{H}_\phi^N[f] = \frac{1}{N} \sum_{j=1}^N H_\phi[f](x_j), \tag{81}$$

and we call the derivatives of this QMC estimate the empirical derivatives of $\mathcal{H}_\phi[f]$. We are interested in proving the convergence of the empirical gradients of $\mathcal{H}_\phi[f]$ with respect to its input $f$ as well as the parameters $\phi$ of the layer. As in Appendix C.1, we write $\pi f = \{f(x_j)\}_{j=1}^N$.

**Theorem 23** *For any DI-Net layer $\mathcal{H}_\phi$ which takes an NF $f \in \mathcal{F}_m$ as input, the empirical gradient of $\mathcal{H}_\phi[f]$ w.r.t. its parameters $\phi$ is convergent in $N$:*

$$\lim_{N \to \infty} \left| \nabla_\phi \hat{H}_\phi^N[f] \right| < \infty. \tag{82}$$

*Additionally, the empirical gradient of $\mathcal{H}_\phi[f]$ w.r.t. its discretized input $\pi f$ is convergent in $N$:*

$$\lim_{N \to \infty} \left| \nabla_{\pi f} \hat{H}_\phi^N[f] \right| < \infty. \tag{83}$$

**Proof**

**NF to Vector: gradients w.r.t. parameters**  Consider the case of a layer $\mathcal{H}_\phi : \mathcal{F}_m \to \mathbb{R}^n$. As in (3), such a layer can be expressed as:

$$\mathcal{H}_\phi[f] = \int_\Omega h(x, f(x), \ldots, D^\alpha f(x); \phi) dx, \tag{84}$$

for some $h$ parameterized by $\phi$, with weak derivatives up to order $|\alpha|$ taken with respect to each channel.

If $\phi = (\phi_1, \ldots, \phi_K)$, then denote $\phi + \tau e_k = (\phi_1, \ldots, \phi_{k-1}, \phi_k + \tau, \phi_{k+1}, \ldots, \phi_K)$.

$$\lim_{N\to\infty} \frac{\partial}{\partial \phi_k} \hat{H}_\phi^N[f] = \lim_{N\to\infty} \frac{\partial}{\partial \phi_k} \left( \frac{1}{N} \sum_{j=1}^N H_\phi[f](x_j) \right) \tag{85}$$

$$= \lim_{N\to\infty} \lim_{\tau\to 0} \left( \frac{1}{\tau N} \sum_{j=1}^N H_{\phi+\tau e_k}[f](x_j) - H_\phi[f](x_j) \right) \tag{86}$$

$$= \lim_{\tau\to 0} \frac{1}{\tau} \int_\Omega h(x, f(x), \ldots, \phi + \tau e_k) - h(x, f(x), \ldots, \phi) dx \tag{87}$$

$$= \lim_{\tau\to 0} \frac{\mathcal{H}_{\phi+\tau e_k}[f] - \mathcal{H}_\phi[f]}{\tau} \tag{88}$$

$$= \frac{\partial}{\partial \phi_k} \mathcal{H}_\phi[f], \tag{89}$$

where (87) follows by eqn. (84) and the Moore-Osgood theorem. Thus the empirical gradient converges to the Jacobian of $\mathcal{H}_\phi$ w.r.t. each parameter, which is finite by assumption.

**NF to NF: gradients w.r.t. parameters**  A DI-Net layer $\bar{\mathcal{H}}_{\phi'} : L^2(\Omega) \to L^2(\Omega)$ can be expressed as:

$$\bar{\mathcal{H}}_{\phi'}[f](x') = \int_\Omega \bar{h}(x, f(x), \ldots, D^\alpha f(x), x', f(x'), \ldots, D^\alpha f(x'); \phi') dx. \tag{90}$$

We can in fact follow the same steps as the NF to Vector case above, to arrive at:

$$\lim_{N\to\infty} \frac{\partial}{\partial \phi_k'} \left( \frac{1}{N} \sum_{j=1}^N \bar{H}_{\phi'}[f] \right) = \frac{\partial}{\partial \phi_k'} \bar{\mathcal{H}}_{\phi'}[f], \tag{91}$$

with equality at each point $x' \in \Omega$ and channel of the output NF.

**NF input: gradients w.r.t. inputs**  Here we combine the NF to vector and NF to NF cases for brevity. For fixed $\tilde{x} \in \Omega$, the empirical derivative of $\mathcal{H}_\phi$ w.r.t. $f(\tilde{x})$ can be written:

$$\frac{\partial}{\partial f(\tilde{x})} \hat{H}_\phi^N[f] = \frac{\partial}{\partial f(\tilde{x})} \left( \frac{1}{N} \sum_{j=1}^N H_\phi[f](x_j) \right) \tag{92}$$

$$= \lim_{\tau\to 0} \frac{1}{\tau N} \sum_{j=1}^N H_\phi[f + \tau \psi_{\tilde{x},N}](x_j) - H_\phi[f](x_j), \tag{93}$$

where $\psi_{\tilde{x},N}$ is any function in $W^{|\alpha|,1}(\Omega)$ that is 1 at $\tilde{x}$ and 0 at every $x_j \neq \tilde{x}$, and whose weak derivatives are 0 at every $x_j$. As an example, take the bump function which vanishes outside a small neighborhood of $\tilde{x}$ and smoothly ramps to 1 on a smaller neighborhood of $\tilde{x}$, making its weak derivatives 0 at $\tilde{x}$.

By (6) we know that the sequences $\left\| \hat{H}_\phi^N[f] - \mathcal{H}_\phi[f] \right\|$ and $\left\| \hat{H}_\phi^N[f + \tau\psi_{\tilde{x},N}] - \mathcal{H}_\phi[f + \tau\psi_{\tilde{x},N}] \right\|$ converge uniformly in $N$ to 0 for any $\tau > 0$, where we can use the $\ell_1$ norm for vector outputs or the $L^1$ norm for NF outputs. So for any $\epsilon > 0$ and any $\tau > 0$, we can choose $N_0$ large enough such that for any $N > N_0$:

$$\left\| \hat{H}_\phi^N[f + \tau\psi_{\tilde{x},N}] - \mathcal{H}_\phi[f + \tau\psi_{\tilde{x},N}] \right\| < \frac{\epsilon}{2}, \tag{94}$$

and

$$\left\| \hat{H}_\phi^N[f] - \mathcal{H}_\phi[f] \right\| < \frac{\epsilon}{2}. \tag{95}$$

Then,

$$\left\| \hat{H}_\phi^N[f + \tau\psi_{\tilde{x},N}] - \mathcal{H}_\phi[f + \tau\psi_{\tilde{x},N}] \right\| + \left\| \hat{H}_\phi^N[f] - \mathcal{H}_\phi[f] \right\| < \epsilon, \tag{96}$$

by the triangle inequality,

$$\left\| (\hat{H}_\phi^N[f + \tau\psi_{\tilde{x},N}] - \mathcal{H}_\phi[f + \tau\psi_{\tilde{x},N}]) - (\hat{H}_\phi^N[f] - \mathcal{H}_\phi[f]) \right\| < \epsilon \tag{97}$$

$$\left\| \left( \hat{H}_\phi^N[f + \tau\psi_{\tilde{x},N}] - \hat{H}_\phi^N[f] \right) - (\mathcal{H}_\phi[f + \tau\psi_{\tilde{x},N}] - \mathcal{H}_\phi[f]) \right\| < \epsilon, \tag{98}$$

hence $\left\| \left( \hat{H}_\phi^N[f + \tau\psi_{\tilde{x},N}] - \hat{H}_\phi^N[f] \right) - (\mathcal{H}_\phi[f + \tau\psi_{\tilde{x},N}] - \mathcal{H}_\phi[f]) \right\|$ converges uniformly to 0. Since the distance between two vectors is 0 iff they are the same, we can write:

$$\lim_{N\to\infty} \hat{H}_\phi^N[f + \tau\psi_{\tilde{x},N}] - \hat{H}_\phi^N[f] = \lim_{N\to\infty} \mathcal{H}_\phi[f + \tau\psi_{\tilde{x},N}] - \mathcal{H}_\phi[f] \tag{99}$$

$$\lim_{\tau\to 0} \frac{1}{\tau} \lim_{N\to\infty} \left( \hat{H}_\phi^N[f + \tau\psi_{\tilde{x},N}] - \hat{H}_\phi^N[f] \right) = \lim_{\tau\to 0} \frac{1}{\tau} \lim_{N\to\infty} (\mathcal{H}_\phi[f + \tau\psi_{\tilde{x},N}] - \mathcal{H}_\phi[f]). \tag{100}$$

By the Moore-Osgood theorem,

$$\lim_{N\to\infty} \lim_{\tau\to 0} \frac{1}{\tau} \left( \hat{H}_\phi^N[f + \tau\psi_{\tilde{x},N}] - \hat{H}_\phi^N[f] \right) = \lim_{N\to\infty} \lim_{\tau\to 0} \frac{1}{\tau} (\mathcal{H}_\phi[f + \tau\psi_{\tilde{x},N}] - \mathcal{H}_\phi[f]) \tag{101}$$

$$\lim_{N\to\infty} \frac{\partial}{\partial f(\tilde{x})} \hat{H}_\phi^N[f] = \lim_{N\to\infty} d\mathcal{H}_\phi[f; \psi_{\tilde{x},N}]. \tag{102}$$

The limit on the right hand side is finite due to Fréchet differentiability of $H_\phi$ at all $f$.

Choose a family $\{\Psi_N\}_{N\in\mathbb{N}}$ of decreasing sequences of bump functions around each $x_j$, $\Psi_N = \{\psi_{x_j,N}\}_{j=1}^N$. An example of such a family is the (appropriately designed) partitions of unity with $N$ elements.

Then the empirical gradient w.r.t. $\pi f$ converges to the limit of the Gateaux derivatives of $\mathcal{H}_\phi$ w.r.t. the bump function sequences in this family. ∎

**Theorem 24** *Chained empirical derivatives converge in $N$ to the chained Gateaux derivatives.*

**Proof** Consider a two-layer DI-Net with NF input $f \mapsto (\mathcal{H}_\theta \circ \mathcal{H}_\phi)[f]$. For the case of derivatives w.r.t. the input, we would like to show the analogue of (102):

$$\lim_{N \to \infty} \frac{\partial}{\partial f(\tilde{x})} \left( \hat{H}_\theta^N \circ \hat{H}_\phi^N \right)[f] = \lim_{N \to \infty} d \left( \mathcal{H}_\theta \circ \mathcal{H}_\phi \right)[f; \psi_{\tilde{x}, N}], \tag{103}$$

where the bump function $\psi_{\tilde{x}, N}$ is defined similarly (1 at $\tilde{x}$ and 0 at each $x_j \neq \tilde{x}$).

$$\frac{\partial}{\partial f(\tilde{x})} \left( \hat{H}_\theta^N \circ \hat{H}_\phi^N \right)[f] = \frac{\partial}{\partial f(x)} \left( H_\theta \left[ \hat{H}_\phi^N[f] \right](x_j) \right) \tag{104}$$

$$= \lim_{\tau \to 0} \frac{1}{\tau} \left( H_\theta \left[ \hat{H}_\phi^N[f + \tau \psi_{\tilde{x}, N}] \right](x_j) - H_\theta \left[ \hat{H}_\phi^N[f] \right](x_j) \right) \tag{105}$$

as in (93).

By (6) we know $\left\| \mathcal{H}_\theta \left[ \hat{H}_\phi^N[f + \tau \psi_{\tilde{x}, N}] \right] - \hat{H}_\theta \left[ \hat{H}_\phi^N[f + \tau \psi_{\tilde{x}, N}] \right] \right\|$ converges to 0 in $N$ for all $\tau > 0$ (where we can use the $\ell_1$ norm for vector outputs or the $L^1$ norm for NF outputs), as does $\left\| \mathcal{H}_\theta \left[ \hat{H}_\phi^N[f] \right] - \hat{H}_\theta \left[ \hat{H}_\phi^N[f] \right] \right\|_{L^1}$. Reasoning as in (94)-(101), we have:

$$\lim_{N \to \infty} \lim_{\tau \to 0} \frac{1}{\tau} \left( \hat{H}_\theta \left[ \hat{H}_\phi^N[f + \tau \psi_{\tilde{x}, N}] \right] - \hat{H}_\theta \left[ \hat{H}_\phi^N[f] \right] \right) \tag{106}$$

$$= \lim_{\tau \to 0} \frac{1}{\tau} \lim_{N \to \infty} \left( \mathcal{H}_\theta \left[ \hat{H}_\phi^N[f + \tau \psi_{\tilde{x}, N}] \right] - \mathcal{H}_\theta \left[ \hat{H}_\phi^N[f] \right] \right) \tag{107}$$

$$= \lim_{N \to \infty} \lim_{\tau \to 0} \frac{1}{\tau} \left( \mathcal{H}_\theta \left[ \mathcal{H}_\phi^N[f + \tau \psi_{\tilde{x}, N}] \right] - \mathcal{H}_\theta \left[ \mathcal{H}_\phi^N[f] \right] \right) \tag{108}$$

Note that

$$d\mathcal{H}[f; \psi_{\tilde{x}, N}] = \frac{1}{\tau} \left( \mathcal{H}_\phi^N[f + \tau \psi_{\tilde{x}, N}] - \mathcal{H}_\phi^N[f] + o(\tau) \right) \tag{109}$$

$$\mathcal{H}_\phi^N[f + \tau \psi_{\tilde{x}, N}] = \mathcal{H}_\phi^N[f] + \tau d\mathcal{H}[f; \psi_{\tilde{x}, N}] + o(\tau). \tag{110}$$

Then we complete the equality in (103) as follows:

$$\text{LHS} = \lim_{N \to \infty} \frac{\partial}{\partial f(\tilde{x})} \left( \hat{H}_\theta^N \circ \hat{H}_\phi^N \right)[f] \tag{111}$$

$$= \lim_{N \to \infty} \lim_{\tau \to 0} \frac{1}{\tau} \left( \mathcal{H}_\theta \left[ \mathcal{H}_\phi^N[f + \tau \psi_{\tilde{x}, N}] \right] - \mathcal{H}_\theta \left[ \mathcal{H}_\phi^N[f] \right] \right) \tag{112}$$

$$= \lim_{N \to \infty} \lim_{\tau \to 0} \frac{1}{\tau} \left( \mathcal{H}_\theta \left[ \mathcal{H}_\phi^N[f] + \tau d\mathcal{H}[f; \psi_{\tilde{x}, N}] \right] - \mathcal{H}_\theta \left[ \mathcal{H}_\phi^N[f] \right] \right) \tag{113}$$

$$= \lim_{N \to \infty} d\mathcal{H}_\theta \left[ \mathcal{H}_\phi[f]; d\mathcal{H}_\phi[f; \psi_{\tilde{x}, N}] \right] \tag{114}$$

$$= \lim_{N \to \infty} d \left( \mathcal{H}_\theta \circ \mathcal{H}_\phi \right)[f; \psi_{\tilde{x}, N}] \tag{115}$$

$$= \text{RHS}, \tag{116}$$

by the chain rule for Gateaux derivatives.

The case of derivatives w.r.t. parameters is straightforward. In the same way we used (94)-(101) to obtain (108), we have:

$$\lim_{N\to\infty} \frac{\partial}{\partial\phi_k}(\hat{H}_\theta^N \circ \hat{H}_\phi^N)[f] = \lim_{N\to\infty}\lim_{\tau\to 0}\frac{1}{\tau}\left(\hat{H}_\theta^N\left[\hat{H}_{\phi+\tau e_k}^N[f]\right] - \hat{H}_\theta^N[\hat{H}_\phi[f]]\right) \tag{117}$$

$$= \lim_{\tau\to 0}\frac{1}{\tau}\left(\mathcal{H}_\theta[\mathcal{H}_{\phi+\tau e_k}[f]] - \mathcal{H}_\theta[\mathcal{H}_\phi[f]]\right) \tag{118}$$

$$= \frac{\partial}{\partial\phi_k}(\mathcal{H}_\theta \circ \mathcal{H}_\phi)[f], \tag{119}$$

Inductively, the chained derivatives extend to an arbitrary number of layers. ■

Since the properties of DI-Net layers extend to loss functions on DI-Nets, we can treat a loss function similarly to a layer, and write:

$$\mathcal{L}_{g'}[g] = \int_\Omega L[g,g'](x)dx \tag{120}$$

$$\hat{L}_{g'}^N[g] = \frac{1}{N}\sum_{j=1}^N L[g,g'](x_j) \tag{121}$$

$$\lim_{N\to\infty}\frac{\partial}{\partial f(\tilde{x})}\left(\hat{L}_{g'}^N \circ \hat{H}_\theta^N \circ \hat{H}_\phi^N\right)[f] = \lim_{N\to\infty} d\left(\mathcal{L}_{g'}\circ\mathcal{H}_\theta\circ\mathcal{H}_\phi\right)[f;\psi_{\tilde{x},N}], \tag{122}$$

where $g'$ can be some other input to the loss function such as ground truth labels. Thus, we can state the following result.

**Corollary 25** *The gradients of a DI-Net's loss function w.r.t. its inputs and all its parameters are convergent in the number of QMC sample points.*

## Appendix D. DI-Net Layers Derived from Grid-Based Networks

We use $c_{\text{in}}$ to denote the number of channels of an input NF and $c_{\text{out}}$ to denote the number of channels of an output NF.

**Convolution** For a measurable $S \subset \Omega$ and a polynomial basis $\{p_{j,\phi}\}_{j\geq 0}$ that spans $L^2(S)$, $S$ is the support of a polynomial convolutional kernel $K_\phi : \Omega \times \Omega \to \mathbb{R}$ defined by:

$$K_\phi(x,x') = \begin{cases} \sum_{j=0}^n p_{j,\phi}(x-x')^j & \text{if } x-x' \in S \\ 0 & \text{otherwise.} \end{cases} \tag{123}$$

for some chosen $n \in \mathbb{N}$. A convolution is the linear map $\mathcal{H}_\phi : L^2(\Omega) \to L^2(\Omega)$ given by:

$$\mathcal{H}_\phi[f] = \int_\Omega K_\phi(\cdot,x')f(x')dx'. \tag{124}$$

An MLP convolution is defined similarly except the kernel becomes $\tilde{K}_\phi(x,x') = \text{MLP}(x-x';\phi)$ in the non-zero case. While MLP kernels are favored over polynomial kernels in many applications due to their expressive power (Wang et al., 2021), polynomial bases can be

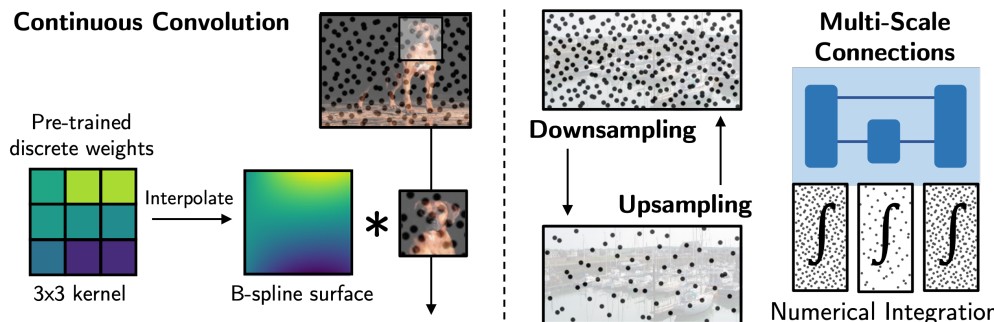

Figure D.2: DI-Nets generalize grid-based networks including convolutional neural networks and vision transformers to arbitrary discretizations of the input and output domains. Low discrepancy point sets used in quasi-Monte Carlo integration are amenable to the multi-scale structures often found in discrete networks. DI-Nets can also be initialized directly from pre-trained discrete networks.

used to construct filters satisfying desired properties such as group equivariance (Cohen and Welling, 2016a,b), $k$-Lipschitz continuity, or boundary conditions.

A multi-channel convolution layer aggregates information locally and across channels. It has $c_{\text{in}}c_{\text{out}}$ learned filters $K_{ij}$, which are defined on some support $S$ which may be a ball or orthotope. The layer also learns scalar biases $b_j$ for each output channel:

$$g_j = \sum_{i=1}^{c_{\text{in}}} K_{ij} * f_i + b_j, \tag{125}$$

with $*$ the continuous convolution as in (124).

To transfer weights from a discrete convolutional layer, $K$ can be parameterized as a rectangular B-spline surface that interpolates the weights (Figure D.2 left). To replicate the behavior of a discrete convolution layer with odd kernel size, $S$ is zero-centered. For even kernel size, we shift $S$ by half the dimensions of a pixel. We use a 2nd order B-spline for $3 \times 3$ filters and 3rd order for larger filters. We use deBoor's algorithm to evaluate the spline at intermediate points.

Different padding behaviors from the discrete case are treated differently. Zero-padding is replicated by scaling $\mathcal{H}[f](x)$ by $\frac{|(S+x)\cap\Omega|}{S+x}$ where $S+x$ is the kernel support $S$ translated by $x$. For reflection padding, the value of the NF at points outside its domain are calculated by reflection. For no padding, the NF's domain is reduced accordingly, dropping all sample points that are no longer on the new domain.

**Multi-scale architectures** Many discretizations permit multi-scale structures by sub-sampling the discretization. Under QMC, downsampling is easily implemented by truncating the list of coordinates in the low-discrepancy sequence to the desired number of terms, as the truncated sequence is itself low-discrepancy. Similarly, upsampling is implemented by extending the low-discrepancy sequence to the desired number of terms, then perform-

ing nearest neighbor interpolation[7]. Since downsampling and upsampling are both specified with respect to the same discretization, DI-Nets can use multi-scale structures with residual or skip connections (Figure D.2 right).

**Linear combinations of channels**   Linear combinations of channels mimic the function of $1 \times 1$ convolutional layers in conventional networks. For learned scalar weights $W_{ij}$ and biases $b_j$:

$$g_j(x) = \sum_{i=1}^{c_{\mathrm{in}}} W_{ij} f_i(x) + b_j, \tag{126}$$

for all $x \in \Omega$. These weights and biases can be straightforwardly copied from a $1 \times 1$ convolutional layer to obtain the same behavior. One can also adopt a normalized version, sometimes used in attention-based networks:

$$W_{ij} = \frac{w_{ij}}{\sum_{k=1}^{c_{\mathrm{in}}} w_{kj}} \tag{127}$$

**Normalization**   All forms of layer normalization readily generalize to the continuous setting by estimating the statistics of each channel with QMC integration, then applying point-wise operations. These layers typically rescale each channel to have some mean $m_i$ and standard deviation $s_i$.

$$\mu_i = \int_\Omega f_i(x) dx \tag{128}$$

$$\sigma_i^2 = \int_\Omega f_i(x)^2 dx - \mu_i^2 \tag{129}$$

$$g_i(x) = \frac{f_i(x) - \mu_i}{\sigma_i + \epsilon} \times s_i + m_i, \tag{130}$$

where we assume $dx$ is normalized and $\epsilon > 0$ is a small constant. Just as in the discrete case, $\mu_i$ and $\sigma_i^2$ can be a moving average of the means and variances observed over the course of training different NFs, and they can also be averaged over a minibatch of NFs (batch normalization) or calculated per datapoint (instance normalization). Mean $m_i$ and standard deviation $s_i$ can be learned directly (batch normalization), conditioned on other data (adaptive instance normalization), or fixed at 0 and 1 respectively (instance normalization).

**Max pooling**   Max pooling is only a well-defined DI-Net layer if each NF channel is in $L^\infty(\Omega)$. Assuming this is the case, there are two natural generalizations of the max pooling layer to a collection of points: 1) assigning each point to the maximum of its k nearest neighbors, and 2) taking the maximum value within a fixed-size window around each point. However, both of these specifications change the output's behavior as the density of points increases. In the first case, nearest neighbors become closer together so pooling occurs over smaller regions where there is less total variation in the NF. In the second case, the empirical maximum increases monotonically as the NF is sampled more finely within each window. Because we may want to change the number of sampling points on the fly, both of these behaviors are detrimental.

---

7. Although other types of interpolation on rectilinear grids do not translate directly to a collection of points, we can replicate their behavior at fixed scale using an appropriately designed convolution filter.

If we consider the role of max pooling as a layer that shuttles gradients through a strong local activation, then it is sufficient to use a fixed-size window with some scaling factor that mitigates the impact of changing the number of sampling points. Consider the following simplistic model: assume each point in a given patch of an NF channel is an i.i.d. sample from $\mathcal{U}([-b, b])$. Then the maximum of $N$ samples $\{f_i(x_j)\}_{j=1}^N$ is on average $\frac{N-1}{N+1}b$. So we can achieve an "unbiased" max pooling layer by taking the maximum value observed in each window and scaling it by $\frac{N+1}{N-1}$ (if $N = 1$ or our empirical maximum is negative then we simply return the maximum), then (optionally) multiplying a constant to match the discrete layer.

To replicate the behavior of a discrete max pooling layer with even kernel size, we shift the window by half the dimensions of a pixel, just as in the case of convolution.

**Tokenization**   To tokenize an NF, we choose a finite set of non-overlapping regions $\omega_j \subset \Omega$ of equal measure such that $\cup_j \omega_j = \Omega$. We apply the indicator function of each set to each channel $f_i$. An embedding of each $f_i|_{\omega_j}$ into $\mathbb{R}^n$ can be obtained by taking its inner product with a polynomial function whose basis spans each $L^2(\omega_j)$. To replicate a pre-trained embedding matrix, we interpolate the weights with B-spline surfaces.

**Average pooling**   An average pooling layer performs a continuous convolution with a box filter, followed by downsampling. To reproduce a discrete average pooling with even kernels, the box filter is shifted, similarly to max pooling.

An adaptive average pooling layer can be replicated by tokenizing the NF and taking the mean of each token to produce a vector of the desired size.

**Parametric functions in $L^2(\Omega)$**   There are several choices of parametric functions in $L^2(\Omega)$. Such functions can enable an NF to be modulated by taking its elementwise product with the function, or for global information to be aggregated in a learned manner by taking its inner product with the function. If $\Omega$ is a subset of $[a, b]^d$, one can use a separable basis defined by the product of rescaled 1D Legendre polynomials along each dimension. If $\Omega$ is a $d$-ball, we can use the Zernike polynomial basis. For any $\Omega$, the function can be represented as an MLP.

**Attention layer**   There are various ways to replicate the functionality of an attention layer. Here we present an approach that preserves the domain. For some $d_k \in \mathbb{N}$ consider a self-attention layer with $c_{\text{in}}d_k$ parametric functions $q_{ij} \in L^2(\Omega)$, $c_{\text{in}}d_k$ parametric functions $k_{ij} \in L^2(\Omega)$, and a convolution with $d_k$ output channels, produce the output NF $g$ as:

$$Q_j = \langle q_{ij}, f_i \rangle \tag{131}$$

$$K_j = \langle k_{ij}, f_i \rangle \tag{132}$$

$$V[f] = \sum_{i=1}^{c_{\text{in}}} v_{ij} * f_i + b_j \tag{133}$$

$$g(x) = \texttt{softmax}\left(\frac{QK^T}{\sqrt{d_k}}\right) V[f](x) \tag{134}$$

A cross-attention layer generates queries from a second input NF. A multihead-attention layer generates several sets of $(Q, K, V)$ triplets and takes the softmax of each set separately.

**Activation Layer**  Pointwise nonlinearities are identical to the discrete case. We use rectified linear units (ReLU) for all our models.

**Data augmentation**  Most data augmentation techniques, including spatial transformations, point-wise functions and normalizations, translate naturally to NFs. Furthermore, spatial transformations are efficient and do not incur the usual cost of interpolating back to the grid. This suggests that DI-Nets may benefit from new data augmentation methods such as adding Gaussian noise to the coordinates of QMC sample points.

## Appendix E. NF Classification

---

**Algorithm E.2** Training for Classification

---

**Input:** network $\mathcal{T}_\theta$, NF/label dataset $\mathcal{D}$, classifier loss $\mathcal{L}$, discretization $x$
**For** step $s \in 1 : N_{\text{steps}}$ **do**

- NFs $f_i$, labels $y_i \leftarrow \text{minibatch}(\mathcal{D})$

- Label estimates $\hat{y}_i \leftarrow \mathcal{T}_\theta(f_i; x)$

- Backpropagate $\mathcal{L}(\hat{y}_i, y_i)$ to update $\theta$

**Output:** trained network $\mathcal{T}_\theta$

---

We perform classification on a dataset of 8,400 NFs fit to a subset of ImageNet1k (Deng et al., 2009), with 700 samples from each of 12 superclasses (Engstrom et al., 2019). The 12 superclasses (dog, structure/construction, bird, clothing, wheeled vehicle, reptile, carnivore, insect, musical instrument, food, furniture, primate) are based on the `big_12` dataset (Engstrom et al., 2019), which is in turn derived from the WordNet hierarchy.

Table E.2: NF classifier top-3 accuracy at two resolutions on ImageNet-Big12.

| Model Type | $32 \times 32$ | $64 \times 64$ |
|---|---|---|
| 2-layer CNN | **59.8%** | 54.1% |
| DI-Net-2 (ours) | 56.6% | **55.2%** |
| 4-layer CNN | **65.7%** | 53.5% |
| DI-Net-4 (ours) | 57.4% | **56.4%** |

For each class we train on 500 SIRENs (Sitzmann et al., 2020b) and evaluate on 200 Gaussian Fourier feature networks (Tancik et al., 2020b). Note that training and testing on different NF types is not possible for hypernetwork or modulation-based approaches. We fit SIREN (Sitzmann et al., 2020b) to each image in ImageNet using 5 fully connected layers with 256 channels and sine non-linearities, trained for 2000 steps with an Adam optimizer at a learning rate of $10^{-4}$. It takes coordinates on $[-1, 1]^2$ and produces RGB values in $[-1, 1]^3$. We fit Gaussian Fourier feature (Tancik et al., 2020b) networks using 4 fully connected layers with 256 channels with ReLU activations. It takes coordinates on $[0, 1]^2$ and produces RGB values in $[0, 1]^3$. The average pixel-wise error of SIREN is $3 \cdot 10^{-4} \pm 2 \cdot 10^{-4}$, compared to $1.6 \cdot 10^{-2} \pm 8 \cdot 10^{-3}$ for Gaussian Fourier feature networks. The difference in quality is visible at high resolution, but indistinguishable at low resolution (Fig. E.3).

We train DI-Nets with 2 and 4 MLP convolutional layers (DI-Net-2 and DI-Net-4), as well as convolutional neural networks (CNNs) with equivalent architectures. Each network is trained for 8K iterations with an AdamW optimizer (Loshchilov and Hutter, 2017) and

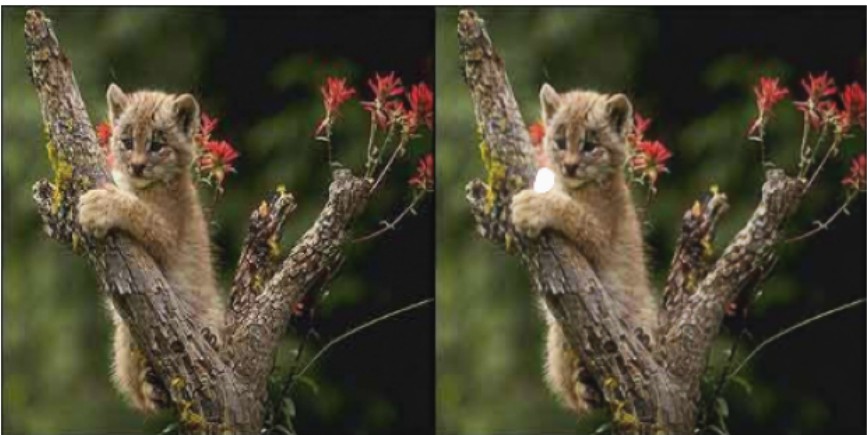

Figure E.3: Comparison of SIREN (left) and Gaussian Fourier feature network (right) representations of an image, rendered at $256 \times 256$ resolution. The Fourier feature network's representation is slightly blurrier (compare the tree bark), but this effect is not noticeable at lower resolutions.

a learning rate of $10^{-3}$. DI-Net-2 uses strided MLP convolutions, a global average pooling layer, then two fully connected layers. DI-Net-4 adds a residual block with two MLP convolutions after the strided convolutions. At training time, the CNNs sample NFs along the $32 \times 32$ grid while DI-Nets sample 1024 points generated from a scrambled Sobol sequence. We evaluate each model with top-3 accuracy at the same resolution as well as at higher resolution ($64 \times 64$ or 4096 points).

DI-Nets somewhat underperform their CNN counterparts (Table E.2), and the gap in performance is larger for the deeper models. However, our discretization invariant model better generalizes to higher-resolution images, where it outperforms CNNs on images at twice the resolution and has a very small drop in performance compared to the resolution it was trained on.

## Appendix F. NF Semantic Segmentation

We perform pixel-wise segmentation on SIRENs fit to street view images from Cityscapes (Cordts et al., 2016), grouping the segmentation labels into 7 categories. SIREN is trained on Cityscapes images for 2500 steps, using the same architecture and settings as ImageNet. The average pixel-wise error is $3.6 \cdot 10^{-4} \pm 1.8 \cdot 10^{-4}$. We train on 2975 NFs with coarsely annotated segmentations only, and test on 500 NFs with both coarse and fine annotations (Fig. F.4). Discrete models sample the $48 \times 96$ grid, and the DI-Nets sample 4608 points. We compare the performance of 3 and 5 layer DI-Nets and fully convolutional networks (FCNs). DI-Net-3 uses three MLP convolutional layers at the same resolution. DI-Net-5 uses a strided MLP convolution to perform downsampling and nearest neighbor interpolation for upsampling. There is a residual connection between the higher resolution layers. Networks

---

**Algorithm F.3** Training for Dense Prediction

---

**Input:** network $\mathcal{T}_\theta$, NF/label dataset $\mathcal{D}$ with dense coordinate-label pairs, task-specific loss $\mathcal{L}$

**For** step $s \in 1 : N_{\text{steps}}$ **do**

- NFs $f_i$, coordinates-label pairs $(x_{ij}, y_{ij}) \leftarrow \text{minibatch}(\mathcal{D})$

- Output NFs $g_i \leftarrow \mathcal{T}_\theta[f_i]$

- Point label estimates $\hat{y}_{ij} \leftarrow g_i(x_{ij})$

- Backpropagate $\mathcal{L}(\hat{y}_{ij}, y_{ij})$ to update $\theta$

**Output:** trained network $\mathcal{T}_\theta$

---

are trained for 10K iterations with a learning rate of $10^{-3}$. We evaluate each model based on mean intersection over union (mIoU) and pixel-wise accuracy (PixAcc).

Table F.3: NF segmentation performance (Cityscapes (Cordts et al., 2016)).

| Model Type | Coarse Segs | | Fine Segs | |
|---|---|---|---|---|
| | **mIoU** | **PixAcc** | **mIoU** | **PixAcc** |
| 3-layer FCN | 0.409 | 69.6% | 0.374 | 63.6% |
| DI-Net-3 (ours) | **0.471** | **78.5%** | **0.417** | **69.4%** |
| 5-layer FCN | **0.488** | **79.4%** | **0.436** | **72.5%** |
| DI-Net-5 (ours) | 0.443 | 77.7% | 0.394 | 68.4% |

DI-Net-3 outperforms the equivalent FCN, and avoids confusing features such as shadows and road markings (Fig. F.4). However, the performance deteriorates when we make the model deeper with a downsampling and upsampling layer (DI-Net-5), and the FCN surpasses both DI-Nets (Table F.3). This echoes the difficulty in scaling DI-Nets observed in classification, and suggests that additional innovations in architecture or training methods are needed for deeper DI-Nets.

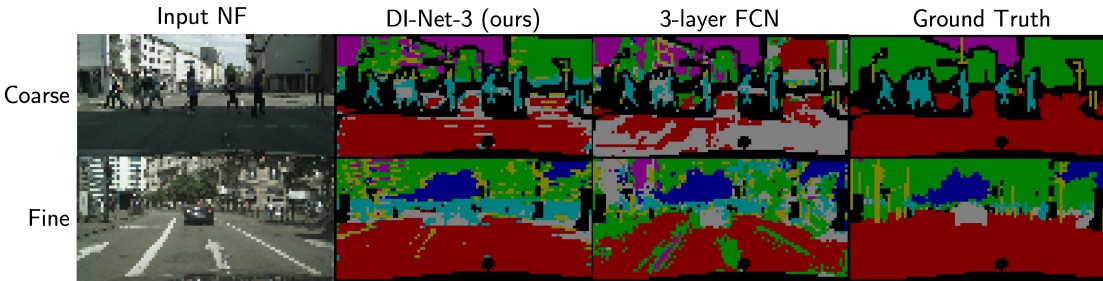

Figure F.4: Cityscapes NF segmentations for models trained on coarse segs only. NF-Net produces NF segmentations, which can be evaluated at the subpixel level.

Table G.4: Pre-trained models fine-tuned on ImageNet NF classification.

| Model Type | Accuracy |
|---|---|
| EfficientNet (Tan and Le, 2019) | **66.4%** |
| DI-Net-EN | 48.1% |

Table G.5: Pre-trained models fine-tuned on Cityscapes segmentation.

| Model Type | Mean IoU | Pixel Accuracy |
|---|---|---|
| ConvNexT (Liu et al., 2022) | **0.429** | 68.1% |
| DI-Net-CN | 0.376 | **68.7%** |

## Appendix G. Analysis

### G.1. Initialization with discrete networks

When a DI-Net is initialized with a large pre-trained discrete network, its outputs are similar[8] by construction. However, the behavior of the pre-trained network is not preserved when the DI-Net switches to QMC – even tiny perturbations from the regular grid are sufficient to change a classifier's predictions. Although the effect on the output of a single layer is much lower than the signal, small differences in each layer accumulate to exert a large influence on the final output. In addition, we find that once grid sampling is abandoned, large DI-Nets cannot easily be fine-tuned to restore the behavior of the discrete network used to initialize it. This suggests that the optimization landscape of DI-Nets may differ substantially from that of discrete networks, and perhaps the approach for learning maps between $L^2$ functions requires fundamentally different strategies than those for learning maps between discrete functions, even when the functions represent the same underlying signals.

Figure G.6 illustrates that the behavior of a pre-trained DI-Net is not preserved when its sampling scheme switches to QMC. For a DI-Net initialized with a truncated EfficientNet, its outputs deviate rapidly as points shift from a regular grid to a low discrepancy sequence.

In Table G.4 and G.5, we illustrate that DI-Net initialized with a large pre-trained discrete network does not match the performance of the original model when fine-tuned with QMC sampling. We use a truncated version of EfficientNet (Tan and Le, 2019) for classification, and fine-tune for 200 samples per class. For segmentation we use a truncated version of ConvNexT-UPerNet (Liu et al., 2022), fine-tuning with 1000 samples.

We also find that the output of an NF-Net is less stable under changing sampling resolution with a grid pattern (Fig. G.5). While the output of a network with QMC sampling converges at high resolution, the grid sampling scheme has unstable outputs until very high resolution. Only the grids that overlap each other (resolutions in powers of two) produce similar activations.

Our preliminary experience with DI-Nets highlights the need for improved sampling schemes and parameterizations that will allow large continuous-domain neural networks to learn effectively. Stable, scalable methods are needed to realize DI-Nets' full potential for continuous data analysis.

---

8. The accumulation of floating-point errors results in a small but noticeable difference in the final result, even with double precision.

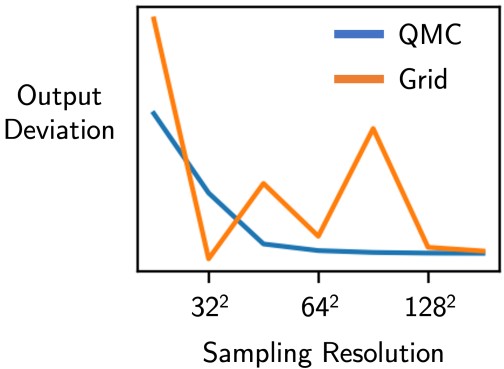

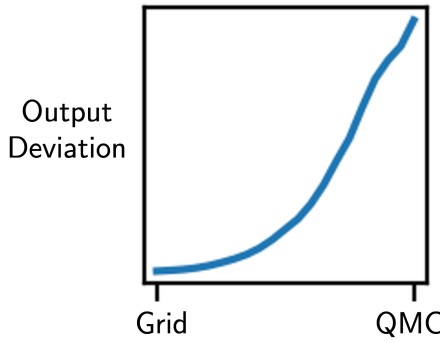

Figure G.5: Distance of the output of a DI-Net from its grid output at $32 \times 32$ resolution, when sampling at various resolutions.

Figure G.6: An DI-Net's output diverges as sample points are gradually shifted from a grid layout to a low discrepancy sequence.

## G.2. Computational Complexity

The DI-Net's complexity is similar to that of a discrete model with an equivalent architecture. In general time and memory both scale linearly with the number of sample points (regardless of the dimensionality of $\Omega$), as well as with network depth and width.

Implemented naively, the computational cost of the continuous convolution is quadratic in the number of sample points, as it must calculate weights separately for each neighboring pair of points. We can reduce this to a linear cost by specifying a $N_{\text{bin}}$ Voronoi partition of the kernel support $B$, then using the value of the kernel at each seed point for all points in its cell. Thus the kernel need only be evaluated $N_{\text{bin}}$ times regardless of the number of sample points. Additionally $N_{\text{bin}}$ can be modified during training and inference.

DI-Net-4 performs a forward pass on a batch of 48 images in $96 \pm 4$ms on a single NVIDIA RTX 2080 Ti GPU.

## Appendix H. Future Directions

**Scaling convolutional DI-Nets**   Our initial experiments suggest that convolutional DI-Nets do not scale well in depth. We suspect that within a CNN-like architecture, the gradients of discrete convolutional layers with respect to kernel parameters have much smoother optimization landscapes over large networks relative to continuous convolutional layers parameterized with MLPs or coefficients of a polynomial basis. It is then no surprise that implementations of neural networks with continuous convolutions do not simply substitute the convolutional layers in a standard CNN architecture, but also make use of a variety of additional techniques (Qi et al., 2017; Wang et al., 2021; Boulch, 2019) which would likely be helpful for scaling convolutional DI-Nets.

**Parameterization of output NFs**   In this work we assume that a DI-Net that produces an NF specifies the output discretization *a priori*, but some applications may need the

output to be sampled several times at different discretizations. It is inefficient to re-evaluate the entire network in such cases, and we propose two solutions for future work. One method can store the last few layers of the network alongside the input activation, and adapt the discretizations as needed in these last few layers only. A second approach can treat the discretized outputs of DI-Net as parameters of the output NF in the manner of Vora et al. (2021), which would maintain interoperability of the entire framework.

**Extending DI-Net to high discrepancy sequences**  In many applications, there are large regions of the domain that are less informative for the task of interest. For example, most of the information in 3D scenes is concentrated at object surfaces, so DI-Nets should not need to process a NeRF by densely sampling all 5 dimensions. Moreover, ground truth labels for dense prediction tasks may only be available along a high discrepancy discretization. Such a discretization can be handled by quadrature, but more work is required to design efficient quadrature methods within a neural network. Additional techniques such as learned coordinate transformations or learned discretizations may also be helpful for extending our model to extreme discretizations or highly non-uniform measures.

**Error propagation**  When an NF does not faithfully represent the underlying data, it is important to characterize the influence on DI-Net's output. In the worst case, these deviations are adversarial examples, and robustness techniques for discrete networks can also be applied to DI-Net. But what can we say about typical deviations of NFs? Future work should analyze patterns in the mistakes that different types of NFs make, and how to make DI-Nets robust to these.

