# OpenReview forum: "Approximate Discretization Invariance for Deep Learning on Neural Fields"
_NeurIPS.cc/2022/Workshop/NeurReps — NeurReps 2022 Poster_

### Official Review · Reviewer_FULT · 2022-10-14
**Discretization Invariant Learning on Neural Fields**

**Confidence:** 1
**Soundness:** 1
**Presentation:** 1
**Contribution:** 1
**Overall Rating:** 3

**Summary:**

 The authors define Nef-NET, a neural network which is invariant with respect to the discretization of the Neural field input. They state an Universal Approximation theorem and they test their result on an image classification task with different discretization methods.

**Questions:**

Which are the different discretization? Do the discretization change from training to test? I expect so as it should be one of the goal of an invariant classifier up to discretization.

**Limitations:**

The authors shortly address the limitation in one sentence which is quite cryptic at the end of the experiment session. What can we actually detect from the experiments?

**Recommended Decision:**

2: Borderline

**Relevance:**

3: Solid fit

**Strengths And Weaknesses:**

The paper addresses an important problem: the invariance with respect to a discretization procedure. However, a 30 pages appendix is quite an absurd as the paper should be read and understandable in 4 pages. The paper is not easy to read in such short number of pages and it is not very clear to me. In addition, the title is about invariance with respect to discretization but the results are about approximation (which is a more reasonable result with respect to the objective, as being invariant under discretization is a very hard task). So the title and the content are not aligned for my understanding. The authors talk also about prediction problems that relates to equivariance better then invariance and should be differently addressed.

Some small comments:

Introduction:
Second paragraph is unclear and should be better written.

Proposition 2. It is not clear to me why the authors only talk about invariance and not about equivariance. If they also cover the case of in-style transfer, should the author talk about equivariance with respect to the discretization. Am I misunderstanding something?

Note 3 should be integrated in the test as it is important in my view.

I don’t understand why the authors talk about approximation. Being invariant with respect to a parametrization means that given a sampling strategy on \Omega, the map from \Omega to \mathbb{R}^c is invariant with respect to the point chosen on \Omega. If the result is a on Approximation (Theorem 5), I think the work should be frames in this direction and not in terms of invariance. The result is still valuable but less strong then the invariance property wrt the discretization.



**Submission Track:**

Extended Abstract (4 Page)

---

### Official Review · Reviewer_tKA2 · 2022-10-15
**Discretization Invariant Learning on NFs**

**Confidence:** 4
**Soundness:** 2
**Presentation:** 2
**Contribution:** 3
**Overall Rating:** 5

**Summary:**

This works aims to address some limitations of training neural networks over neural fields. The goal is to design a neural network, called NF-Net, that is agnostic to the type of neural field and invariant to the data discretization.
An NF-Net layer is defined to be a mapping from neural fields to neural fields (dense prediction), from neural fields to vectors (classification), or from vectors to neural fields (generative model).
Several basic operators in grid-based neural networks, such as convolution, normalization and pooling, are extended to NF-Net layers which are applicable to any type of neural fields.
Relying on Koksma-Hlawka inequality, functionals with the integral form, under mild assumptions, have bounded deviations from their discretized counterparts, provided that the discretization is of low-discrepancy (which can be achieved using quasi-MC). Invariance to discretization is defined accordingly and it is shown (in the appendix) that NF-Nets can approximate any Lipschitz continuous map. For evaluation, classifiers are trained on neural fields to label images. Some further results are also provided for dense prediction in the appendix.

**Questions:**

* Questions:
  1. Are the accuracies reported in table 1 low compared to the accuracy of grid-based networks?
  2. Can you please compare your work with baselines of hypernetworks and modulation-based methods? Do they trade interoperability with better final performance? If so, then is it worth to gain interoperability in the expense of degrading of accuracy?
  3. Have you investigated if NF-Nets underfit or overfit? What is the difference between the train and test accuracy on the classification task? Is the training accuracy also low compared to that of grid-based networks?
  4. It is shown that NF-Nets are universal approximators of large class of functionals over neural fields. With that said, as this is a new kind of network, do you experience any difficulty in their optimization to the final solution?

* Suggestions:
  1. Although a thorough analysis of Koksma-Hlawka inequality is provided in the appendix, it would have been better to add some high-level summary in the main text describing the close relation between inequality and Definition 3.
  2. NF-Net layer is mainly discussed in the main text in the abstract sense. However, realization of such layers can be considered as the contribution of your work whereas it is only discussed in the supplementary material. If I were you, I would have provided at least a brief description for one of the examples, like the convolution, for the sake of better understanding of a typical NF-Net layer. Given the restriction over the space, I would have reduced some less important technical details to fit this new section.
  3. This point might be a philosophical one: Invariance is itself a strict term as it implies absolute zero variance; However, there could be (yet bounded) difference between the functional and its induced map under given points. I suggest that you pick another term that fits better with this property of bounded variance.

**Limitations:**

Despite the approximation guarantees for NF-Nets, it is not clear if it is working well in practice or not. More empirical evidence is required to inspect NF-Nets underfit or overfit and decide if they are practical.

**Recommended Decision:**

2: Borderline

**Relevance:**

2: Limited relevance

**Strengths And Weaknesses:**

* Strengths:
  1. This work introduces a novel framework, different from hypernetworks or modulations-based ones, applicable to a broad range of tasks including but not limited to classification, generation and dense prediction. It opens a new direction to process neural fields as a continuous representation of data in a variety of tasks.
  2. This framework has a theoretical foundation and a comprehensive set of layers are tailored to be used in NF-Net.

* Weaknesses:
  1. The Definition 3 and Koksma-Hlawka inequality are highly relevant, but their relation is not discussed in the main text, which makes it unclear at first sight why this definition implies integral forms for discretization invariant maps.
  2. Although design of NF-Net layers requires great effort and is an important contribution of this work, unfortunately no section is dedicated to it in the main text.
  3. There is a lack of comparison with baselines of hypernetworks or modulation-based methods for the classification task.
  4. There is no discussion over the limitations.

**Submission Track:**

Extended Abstract (4 Page)

---

### Official Review · Reviewer_hF6X · 2022-10-18
**Looks like a solid contribution that could be more widely accessible with a few tweaks on the intuitions behind provided definitions and results**

**Confidence:** 2
**Soundness:** 3
**Presentation:** 2
**Contribution:** 3
**Overall Rating:** 6

**Summary:**

The paper aims to build a framework for training neural networks on neural field (NF) encoded data that (1) does not carry the interpolation errors of first converting the NF data into pixels before applying standard neural nets and (2) is interoperable between different types of NFs.

To do so, the authors introduce a framework (termed “NF-Net”) for building discretisation invariant neural networks for learning on neural fields. The framework is built on leveraging numerical integration.

**Questions:**

*Disclaimer*: I am not an expert in this field.

[Q1] It is still not clear to me whether the resulting discretisation achieved is *exact* or *approximate*? My understanding is that it is approximate but bounded, but I might be wrong? It would be great if the authors could add a discussion on this to the paper.

**Limitations:**

A good discussion of limitation is available in the appendix.

**Recommended Decision:**

3: Accept

**Relevance:**

3: Solid fit

**Strengths And Weaknesses:**

[S1] The paper treats a wide generality of NFs and tasks.
[S2] The paper augments theoretical work with empiricial results (see supplementary)

[W1] As someone who is not an expert in this area, I struggled to understand the main section, despite augmenting my reading with online resources and the supplementary materials. To improve clarity and accessibility, I suggest the authors add intuition to their formal definitions. This could come in the form of (counter)examples or illustrations, to provide the reader with the relevant intuition. I expect that this could greatly increase the impact of this work.
[W2] The organisation of the paper was somewhat confusing to me. In particular, page 4 read mostly like a guide to the various appendix sections rather than a mostly self-contained discussion with reference to the appendix for details. I appreciate that this might be the effect of the 4 page workshop limit for this rather extensive paper.

**Submission Track:**

Extended Abstract (4 Page)

---

### Decision · Program_Chairs · 2022-10-21

Accept (Poster)